# Molecular mechanism of mitochondrial phosphatidate transfer by Ups1

Jiuwei Lu[1,2,6], Chun Chan [3,5,6], Leiye Yu [1,2,6], Jun Fan [3,4✉], Fei Sun [1,2✉] & Yujia Zhai[1✉]

Cardiolipin, an essential mitochondrial physiological regulator, is synthesized from phosphatidic acid (PA) in the inner mitochondrial membrane (IMM). PA is synthesized in the endoplasmic reticulum and transferred to the IMM via the outer mitochondrial membrane (OMM) under mediation by the Ups1/Mdm35 protein family. Despite the availability of numerous crystal structures, the detailed mechanism underlying PA transfer between mitochondrial membranes remains unclear. Here, a model of Ups1/Mdm35-membrane interaction is established using combined crystallographic data, all-atom molecular dynamics simulations, extensive structural comparisons, and biophysical assays. The α2-loop, L2-loop, and α3 helix of Ups1 mediate membrane interactions. Moreover, non-complexed Ups1 on membranes is found to be a key transition state for PA transfer. The membrane-bound non-complexed Ups1/ membrane-bound Ups1 ratio, which can be regulated by environmental pH, is inversely correlated with the PA transfer activity of Ups1/Mdm35. These results demonstrate a new model of the fine conformational changes of Ups1/Mdm35 during PA transfer.

[1] National Laboratory of Biomacromolecules, CAS Center for Excellence in Biomacromolecules, Institute of Biophysics, Chinese Academy of Sciences, 100101 Beijing, China. [2] School of Life Sciences, University of Chinese Academy of Sciences, 100049 Beijing, China. [3] Department of Materials Science and Engineering, City University of Hong Kong, Hong Kong, China. [4] Center for Advanced Nuclear Safety and Sustainable Development, City University of Hong Kong, Hong Kong, China. [5] Present address: College of Pharmacy, The Ohio State University, Columbus, OH 43210, USA. [6] These authors contributed equally: Jiuwei Lu, Chun Chan, Leiye Yu. ✉email: junfan@cityu.edu.hk; feisun@ibp.ac.cn; yujia@ibp.ac.cn

Mitochondria generate energy through oxidative phosphorylation, where ATP is generated through the electron transfer chain and ATP synthase embedded in the inner mitochondrial membrane (IMM). The IMM contains high cardiolipin (CL) levels, accounting for ~20% of the IMM lipid content[1]. CL interacts with numerous IMM proteins and serves as an integral component of respiratory complexes and participates in their folding[2]. The role of CL in mitochondrial signaling pathways has also been emphasized. On receiving stress signals, CL is externalized to the outer mitochondrial membrane (OMM) forming a binding platform, where specifically recruited signaling molecules bind to induce mitophagy and apoptosis[3]. In eukaryotes, CL is synthesized at the matrix side of the IMM via an enzymatic cascade initiating with phosphatidic acid (PA)[4]. PA is a rare mitochondrial membrane component (<1% of the total lipids) and is predominantly synthesized in the endoplasmic reticulum and imported to the OMM[1,5]. Although it remains unclear how PA is transferred from the endoplasmic reticulum to the OMM, a heterodimeric protein complex termed Ups1/Mdm35 in yeast or PRELID1/TRIAP1 in mammalian cells has been identified as a lipid transfer complex that mediates the transport of PA from the OMM to the IMM[6–10].

Ups1 is located in the mitochondrial intermembrane space and is a member of the conserved UPS1/PRELI protein family, which is associated with mitochondrial function[11]. Ups1 was initially reported to affect biogenesis of Mgm1 (a human OPA1 homolog required for mitochondrial fusion)[12]. Ups1-deficient cells have low short-form Mgm1 (s-Mgm1) levels and less tubular mitochondria[12]. Ups1 deletion reportedly decreases CL levels[13]. However, this reduction was restored through simultaneous Ups2 (a yeast Ups1 homolog) deletion. Mdm35 is another conserved yeast intermembrane space protein belonging to the twin Cx9C protein family[14]. As the binding partner of the Ups proteins, Mdm35 facilitates their mitochondrial import and protects them against proteolysis[15,16]. The role of the Ups1/Mdm35 complex in lipid transfer was first reported in 2012; thereafter, Ups1 was recognized as a lipid transfer protein mediating PA shuttling between mitochondrial membranes[8]. Furthermore, the dynamic assembly of Ups1 with Mdm35 was found to be essential for PA transfer. The closest homolog of Ups1/Mdm35 in human is PRELID1/TRIAP1, which also facilitates PA transfer in vitro[9]. Along with PRELID1, humans possesses three other Ups1 homologs named PRELID2, SLMO1, and SLMO2, all sharing a conserved "PRELID" domain with PRELID1 and forming a complex with TRIAP1 (a Mdm35 homolog)[10].

In 2015, two groups reported the crystal structures of Ups1/Mdm35 with or without PA, revealing Ups1–Mdm35 interactions and the PA-binding pocket of Ups1[17,18]. Simultaneously, Miliara et al. reported the crystal structure of SLMO1/TRIAP1, structurally similar to Ups1/Mdm35. Despite no obvious sequence homology, Ups1/Mdm35 displays remote structural similarity to other lipid transfer proteins including phosphatidylinositol transfer proteins (PITPs) and the related cholesterol-binding START domains[19,20]. Structural comparisons revealed that the α2-loop in Ups1 may be the equivalent of the lipid exchange loop of PITPα or the Ω1-loop in the START domains[17,21]. These two loops are considered membrane-docking sites for lipid loading and unloading, and function as lids of the lipid-binding cavity. Outward and inward lid movements expose and conceal the cavity entrance, respectively, facilitating lipid entry/exit from the cavity or their retention in the cavity[19,22]. In the apo- and lipid-bound PITPα structure, this lid is differently positioned, resulting in distinct opened and closed conformations of the lipid-binding cavity.

Different mechanisms, however, prevail in the Ups1/Mdm35 system. The α2-loop is critical for Ups1 function, where deleting this loop or replacing all residues with Ala or substituting conserved L62 and W65 to Ala drastically impairs PA transfer by Ups1/Mdm35. However, the α2-loop conformation unexpectedly changes slightly between the apo and PA-bound Ups1/Mdm35 conformations[17,18]. Furthermore, an α2-loop-deficient Ups1 mutant remains bound to CL-bearing liposomes, suggesting that the α2-loop is irrelevant to the membrane-docking site[17]. Mutagenesis studies have reported that some conserved basic residues either at the base of the PA-binding pocket (R25 and R54) or at the periphery near the PA-binding site (H33, K58, K61, K148, and K155) are critical for PA transfer[18,21]. Furthermore, Mdm35 dissociation enhanced Ups1 binding to the membrane[17]. Crystal structures of human PRELID1-TRIAP1 and PRELID3b-TRIAP1 were resolved recently, and the structural determinants of lipid specificity were studied through yeast genetic screens and coarse-grain simulations[23].

These studies improve our understanding of the mechanism underlying PA transfer between mitochondrial membranes. However, additional details regarding PA transfer (e.g., how Ups1/Mdm35 or Ups1 bind to the membrane, how Mdm35 dissociates from Ups1, and the precise role of the α2-loop regarding whether it undergoes a conformational change before and after PA-binding and how PA transfer is regulated) remain elusive. In this study, we combined crystallography, molecular dynamics (MD) simulations and biophysical assays to further elucidate the mechanism underlying Ups1/Mdm35-mediated PA transfer.

## Results

**Structure of monomeric Ups1/Mdm35 complex.** The crystal structure of selenomethionine-substituted *Saccharomyces cerevisiae* Ups1/Mdm35 (PDB code: 5JQL) was determined at a resolution of 2.9 Å through the single-wavelength anomalous dispersion method (Table 1).The asymmetric unit of the crystal contains six Ups1/Mdm35 heterodimers (Supplementary Fig. 1), which are largely similar; however, the C-terminal long α3 helices of Ups1 are somewhat differently positioned (Supplementary Fig. 2).The C-terminal long α3 helices (residues K138–E168) of every two Ups1 (Ups1A and Ups1B) exchange with each other, so the two Ups1 molecules form a domain-swapped dimer (Fig. 1a), concurrent with the previously reported structure (PDB code: 4YTW) by Watanabe et al.[17], whereby the domain-swapped dimer was speculated to originate from crystallization artifacts during gel filtration analysis and cysteine crosslinking assay. For subsequent structural analysis, they generated an artificial Ups1/Mdm35 monomer[17]. However, it is unclear whether this artificial structure represents the native Ups1/Mdm35 monomer.

To obtain the true Ups1/Mdm35 monomer, we generated an Ups1–Mdm35 fusion protein having the same PA transfer activity as the wild-type Ups1/Mdm35 (WT), where a PreScission protease recognition sequence LEVLFQGP was inserted between the Ups1 C-terminal and Mdm35 N-terminal (Fig. 1b). The structure of this Ups1–Mdm35 -protein was resolved (PDB code, 5JQM) and refined to 1.5 Å through molecular replacement, containing three monomeric Ups1–Mdm35 fusion proteins per asymmetric unit (Fig. 1c). The Ups1–Mdm35 linker is invisible, suggesting its high flexibility. Because the three molecules can be accurately superimposed with a root-mean-square deviation (RMSD) of ~0.1 Å, we selected the most complete Ups1 model from chain B and Mdm35 model from chain C to reconstruct the Ups1/Mdm35 structure (referred as UM-fusion in the following context) for further structural analysis (Fig. 1d).

Monomeric Ups1 in UM-fusion forms a "hot-dog" fold through seven-stranded antiparallel β-sheets (β1–β7) and three α helices (α1–α3). Residues between β3 and β4 form the α2-loop

**Table 1 Data collection and refinement statistics.**

| | SeMet-labeled Ups1/Mdm35 | Ups1–Mdm35 | Ups1/Mdm35–DHPA |
|---|---|---|---|
| Data collection | | | |
| Space group | $P2_1$ | $C2$ | $P4_1$ |
| Cell dimensions | | | |
| $a$, $b$, $c$ (Å) | 83.3, 130.7, 84.3 | 118.6, 68.3, 105.1 | 90.1, 90.1, 81.3 |
| $\alpha$, $\beta$, $\gamma$ (°) | 90, 103.1, 90 | 90, 102.1, 90 | 90, 90, 90 |
| Resolution (Å) | 50–2.90 | 50–1.50 | 50–3.36 |
| $R_{sym}$ or $R_{merge}$ | 0.074 (0.438)[a] | 0.084 (0.42) | 0.059 (1.09) |
| $I / \sigma I$ | 31.7(4.2) | 37.7 (3.0) | 9.59 (1.05) |
| Completeness (%) | 98.4 (97.0) | 98.6 (88.9) | 99.8 (99.8) |
| Redundancy | 3.6 (3.6) | 3.6 (3.1) | 6.8 (6.9) |
| Refinement | | | |
| Resolution (Å) | 50–2.90 | 50–1.50 | 50–3.36 |
| No. of reflections | 39,282 | 129,414 | 9395 |
| $R_{work}/R_{free}$ | 20.0/24.8 | 16.4/20.2 | 27.2/29.7 |
| No. of atoms | | | |
| Protein | 11,018 | 5610 | 3308 |
| Ligand/ion | — | — | 44 |
| Water | — | 811 | — |
| B-factors | | | |
| Protein | 38.1 | 23.0 | 149.9 |
| Ligand/ion | — | — | 163.8 |
| Water | — | 35.1 | — |
| R.m.s. deviations | | | |
| Bond lengths (Å) | 0.01 | 0.007 | 0.002 |
| Bond angles (°) | 1.3 | 1.1 | 0.49 |

[a]Values in parentheses are for highest-resolution shell.

(Fig. 1d). UM-fusion is structurally similar to the artificial Ups1/Mdm35 monomers (PDB codes: 5JQL and 4YTW), with RMSDs of 0.704 and 0.766 Å for 239 and 230 Cα atoms, respectively (Fig. 1e). Marked differences were observed in the electron density of residues F133–K138, which is clear and continuous in UM-fusion but lacking in artificial monomers. Mdm35 contains three α helices (αA–αC) with two disulfide bridges (C13–C52 and C23–C42) between αA and αB helices (Fig. 1d). These three α helices embrace Ups1, yielding a 1222 Å$^2$ buried surface area, being ~12.7% of the total Ups1 surface area. The Ups1 PA-binding pocket and Ups1–Mdm35 interactions have been previously described in detail[17,18].

**Structure of DHPA-bound Ups1/Mdm35.** To investigate the mechanism underlying Ups1/Mdm35-mediated PA transfer, we attempted to obtain the structure of PA-bound Ups1/Mdm35. Both Ups1/Mdm35 and Ups-Mdm35 fusion protein were co-crystallized with PA with different acyl chains, including 1,2-dioleoyl-sn-glycero-3-phosphate (DOPA; 18:1–18:1), 1-palmitoyl-2-oleoyl-sn-glycero-3-phosphate (POPA; 16:0–18:1), and 1,2-dihexanoyl-sn-glycero-3-phosphate (DHPA; 6:0–6:0). Although we obtained DOPA- and DHPA-bound Ups1/Mdm35 crystals, only DHPA-bound Ups1/Mdm35 crystals diffracted well. Finally, we determined the structure of Ups1/Mdm35–DHPA complex (PDB code: 6KYL) at 3.35 Å resolution through molecular replacement with UM-fusion as the starting model.

Two Ups1/Mdm35 molecules were present per asymmetric unit in Ups1/Mdm35–DHPA structure (Fig. 2a). The Ups1 molecules also form a domain-swapped dimer. The Mdm35 molecule presents a none-classic conformation, where its αC-helix does not bend toward the αB-helix N-terminal but extends toward the αB-helix. On analyzing the extent of crystal contacts, αC helices (residues 62–74) of two adjacent Mdm35 molecules were exchanged, causing interactions between Mdm35 αC-helix

and the adjacent Ups1 (Supplementary Fig. 3). Ups1/Mdm35–DHPA is structurally similar to the previously reported Ups1/Mdm35–DLPA (PDB code, 4YTX)[17], with an RMSD of 0.86 Å for 155 Cα atoms (Supplementary Fig. 4). Yu et al.[18] reported a monomeric Ups1/Mdm35-PA structure (PDB code: 4XHR). However, after re-examined this structure, N134–I137 poorly fit with the density map; hence, we reconstructed this part of the structure and found that two Ups1 molecules form a domain-swapped dimer (Supplementary Fig. 5).

In Ups1/Mdm35–DHPA structure, each Ups1/Mdm35 contains a DHPA molecule with high real-space correlation coefficients (Supplementary Fig. 6). Residues V66–G72 at the α2-loop could not be assigned because of a low electron density, suggesting the high flexibility of this loop. The phosphate head of bound DHPA is proximal to the positively charged R25 side chain, and the H33, K58, and N152 side chains are adequately proximal to facilitate electrostatic interactions or hydrogen bonds with the phosphate oxygen atoms. The sn-1 and sn-2 acyl chains of DHPA were stabilized via hydrophobic interactions with T76, I78, M104, and V106 (Fig. 2b). Structural comparison of Ups1/Mdm35–DLPA and Ups1/Mdm35–DHPA revealed that DLPA is inserted deeper into the positively charged pocket than DHPA (Supplementary Fig. 7).

**A new Ups1/Mdm35 interaction mode.** In previously reported Ups1/Mdm35 structures, all Mdm35 helices interact with the same Ups1 molecule, constituting the classic Ups1–Mdm35 interaction mode (Fig. 2c). However, in Ups1/Mdm35–DHPA, Mdm35 αA and αB helices interact with a Ups1 molecule, whereas its αC-helix interacts with another Ups1 molecule. The interaction area between this αC-helix and its bound Ups1 is ~375 Å$^2$, being slightly smaller than the interaction area (~431 Å$^2$) of Mdm35 αC-helix–Ups1 in the classic interaction. Although this new interaction mode could be a crystallization artifact, it reflects αC-helix flexibility.

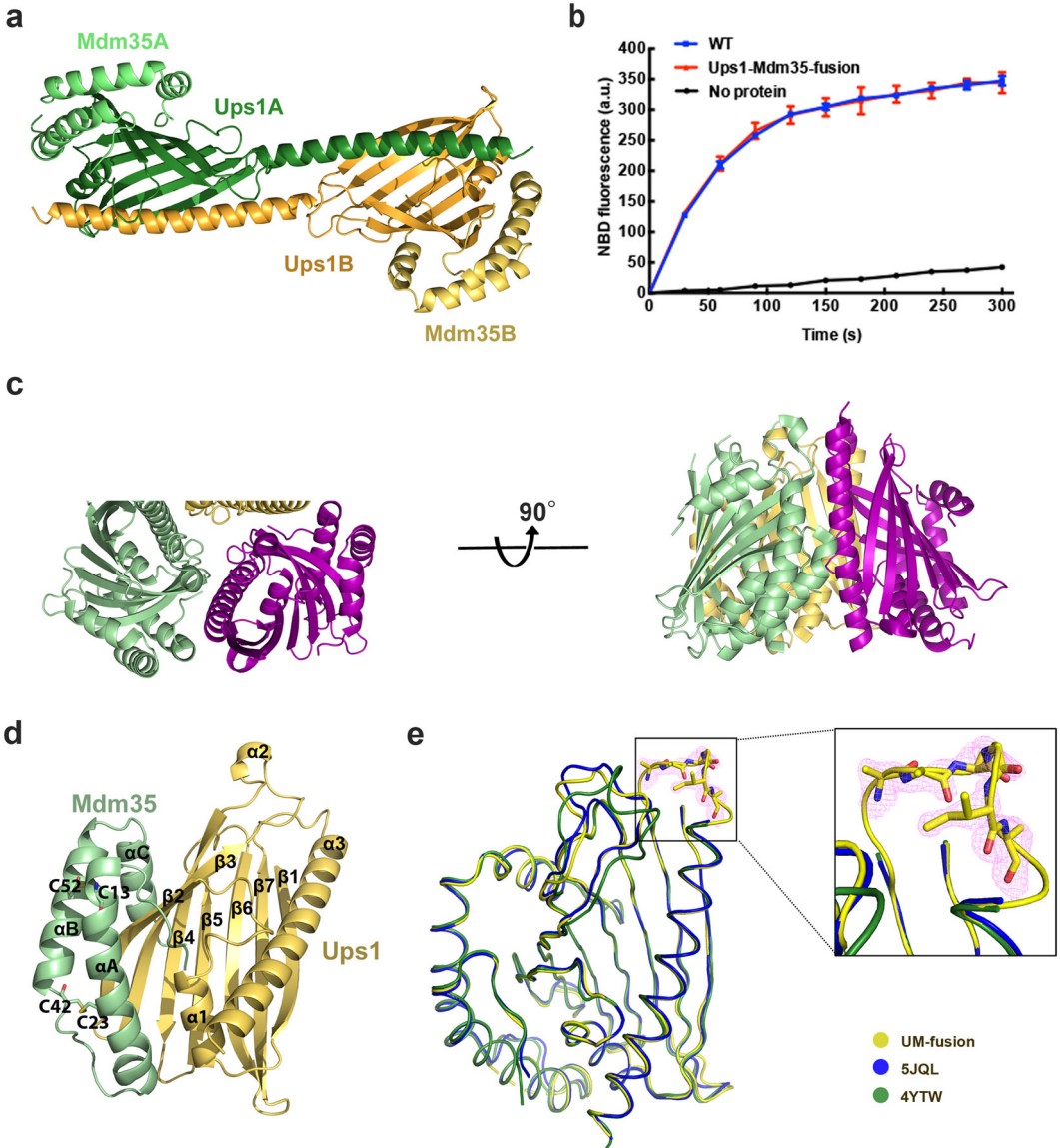

**Fig. 1 Crystal structures of the Ups1/Mdm35 complex and Ups1–Mdm35 fusion protein (UM-fusion). a** The domain-swapped dimer of Ups1/Mdm35 (PDB code: 5JQL). Ups1A, Ups1B, Mdm35A, and Mdm35B are colored in forest, orange, light green, and light yellow, respectively. **b** PA transfer activities of Ups1/Mdm35 and UM-fusion. Traces indicate the mean ± s.d. values from three independent experiments. **c** A trimer formed by three UM-fusion proteins in an asymmetric unit (PDB code: 5JQM). **d** Ribbon diagram of UM-fusion. The Ups1 part is colored in light yellow and the Mdm35 part is colored in light green. The secondary structures are labeled accordingly. **e** Superposition of Ups1/Mdm35 structures (PDB codes: 5JQM, 5JQL, and 4YTW). Magnified view shows the $2F_o - F_c$ omit map (contoured at 1.0 $\sigma$) of the L8-Loop of Ups1 in UM-fusion.

To determine whether Mdm35 αC-helix affects Ups1/Mdm35-mediated PA transfer, we generated three Ups1/Mdm35 truncation mutants (Fig. 2d) for functional assays. The Ups1/Mdm35ΔC5 mutant lacked five Mdm35 C-terminal residues (K82–K86) and was used as a negative control because this mutant was reported to have the same PA transfer activity as full-length Ups1/Mdm35[17]. The Ups1/Mdm35ΔC14 mutant lacked 14 Mdm35 C-terminal residues (A73–K86), which were untraceable in the structure but located immediately after the αC-helix. Secondary structure prediction revealed that this segment is disordered. In the Ups1/Mdm35Δ(αC + C14) mutant, the Mdm35 αC-helix was further deleted, thus only Mdm35 αA and αB helices interact with Ups1. For brevity, these three mutants were abbreviated as ΔC5, ΔC14, and Δ(αC + C14). Ups1–Mdm35 interactions in these mutants were confirmed through gel filtration (Supplementary Fig. 8a). The thermal shift assay revealed that the

critical temperatures were much lower in ΔC14 and Δ(αC + C14) than in WT, indicating reduced Ups1–Mdm35 interactions in these mutants (Supplementary Fig. 8b, c). We measured PA transfer activity in these mutants through a fluorescence-based PA transfer assay (Fig. 2e, f). ΔC5 and ΔC14 truncations had almost no effects on NBD fluorescence elevation, whereas Δ(αC + C14) truncation considerably impaired PA transfer activity. Hence, interactions between Mdm35 αC-helix and Ups1 are essential for PA transfer.

To determine whether their interactions would affect the membrane-binding ability of Ups1, we performed liposome co-sedimentation assays (Fig. 2g). When a system comprising Ups1/Mdm35 and PA-containing liposomes attains equilibrium, the following types of molecules should be present: first, a liposome-bound Ups1/Mdm35 (state A), noted because Mdm35 itself does not bind to liposomes, whereas liposome-bound Mdm35 has been

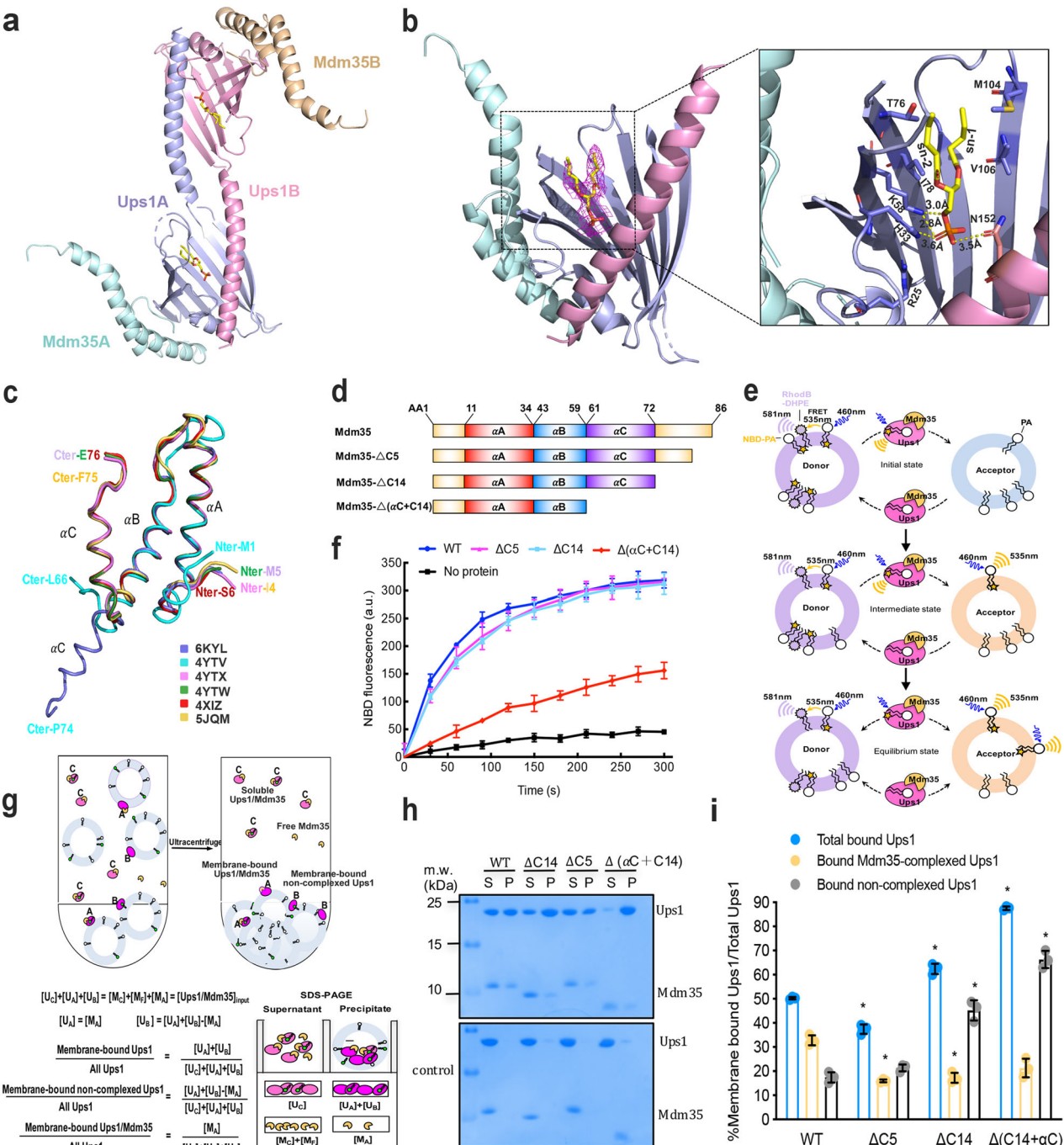

**Fig. 2 Structure of Ups1/Mdm35–DHPA, PA transfer activity, and the lipid co-sedimentation assay. a** The domain-swapped dimer of Ups1/Mdm35–DHPA in an asymmetric unit (PDB code: 6KYL). Ups1 is denoted in pink and light purple. Mdm35 is colored in wheat and pale cyan. The DHPA molecule is shown with a stick model with carbon atoms colored in yellow; oxygen atoms, red; phosphorus atoms, orange. **b** Binding of DHPA in the pocket. The simulated annealing $F_o - F_c$ map was generated by omitting the DHPA molecule, and is denoted in magenta meshes countered at 2.0 σ. A magnified view displaying the residues around the DHPA molecule. Dotted lines indicate potential hydrogen bonds. **c** Superposition of Mdm35 structures. **d** Schematic diagrams of Mdm35 and its truncation mutants. **e** Schematic representation of the fluorescence-based PA transfer assay. **f** PA transfer activities of the WT, ΔC5, ΔC14, and Δ(αC + C14). Traces display mean ± s.d. values of three independent experiments. **g** Schematic representation of the liposome co-sedimentation assay. The formula is used to determine the percentage of Ups1 under different liposome-binding states. **h** Representative Tricine-SDS-PAGE analysis of liposome co-sedimentation assays of the WT, ΔC5, ΔC14, and Δ(αC + C14) from at least three independent experiments. The raw images of gels are provided in Supplementary Fig. 14. **i** Quantification of Ups1 in different liposome-binding states. The Ups1 and Mdm35 signals on gels (**h**) were quantified and their values were substituted into the formula displayed in **g**. The data represent three independent experiments, and values are expressed as mean ± s.d. values. An independent samples t-test was used for statistical analyses via SPSS (version 23.0; n = 3, *p < 0.01).

detected in a series of previously reported Ups1/Mdm35 liposome flotation assays[8,21]; second, a membrane-bound non-complexed Ups1 (state B), which was previously detected in liposome flotation assays[8,17,21]; and finally, a cytosolic Ups1/Mdm35 (state C) and Mdm35 that dissociates from liposome-binding Ups1. Soluble proteins and liposome-bound proteins were separated through ultracentrifugation. The resulting supernatant and precipitate fractions were analyzed through sodium dodecyl sulfate (SDS) polyacrylamide gel electrophoresis (PAGE; Fig. 2h). On the basis of the band intensities of Ups1 and Mdm35 in the two fractions, Ups1 distribution in states A and B were determined.

The overall membrane-binding ability of $\Delta(\alpha C + C14)$ mutant was significantly higher than that of its WT counterpart (Fig. 2i), primarily because of considerably increased liposome-bound non-complexed Ups1, indicating that weakening of Ups1–Mdm35 interactions retains more non-complexed Ups1 at the membrane. Interestingly, the amount of membrane-bound non-complexed Ups1 in $\Delta$C14 mutant also significantly increased, suggesting that 14 C-terminal Mdm35 residues potentially interact with Ups1, although the high flexibility of this segment makes it invisible in all crystal structures.

**Ups1–Mdm35 membrane interactions**. The liposome co-sedimentation assay revealed that both Ups1/Mdm35 and non-complexed Ups1 bound to the membrane. To investigate how Ups1 interacts with the membrane with or without Mdm35, we performed all-atom MD simulations where either a freestanding Ups1 (Ups1$_{free}$) or an Ups1/Mdm35 complex interacts with DOPA-containing lipid bilayers (Supplementary Movies 1 and 2). Because the Ups1 $\alpha$2-loop was considered an equivalent of the lipid exchange loop of PITP$\alpha$, it may interact with the membrane in the same orientation as the lipid exchange loop. Thus, we selected the initial orientation of Ups1 to make the long axis of the $\alpha$2-loop perpendicular to the membrane plane (Fig. 3a). In total, 1040 and 600 ns trajectories were generated for Ups1$_{free}$ and Ups1/Mdm35 systems, respectively. The RMSD of Ups1$_{free}$ was relatively small (<3 Å) and converged sufficiently after 300 ns (Fig. 3b). At ~400 ns, Ups1$_{free}$ started interacting with DOPA-containing lipid bilayers in a stable orientation and showed no significant structural changes, evident from converged RMSD profiles and a sharp reduction in protein–lipid separation (Fig. 3c). In contrast, the Ups1/Mdm35 complex showed relatively large RMSD for both Ups1 (Ups1$_{complex}$) and Mdm35 (Mdm35$_{complex}$) parts, and did not begin to converge until 500 ns (Fig. 3b). During the first 400 ns of the simulation, the Ups1/Mdm35 complex randomly interacted with the membrane and showed a relatively large RMSD (~3 Å) (Supplementary Movie 2). After 400 ns, although it interacted with the membrane in an orientation similar to that of Ups1$_{free}$, the protein complex deviated further with a larger RMSD (>3 Å), which was counter-intuitive, as the presence of Mdm35$_{complex}$ was considered to stabilize residues at the protein–protein interface, thus decreasing the overall RMSD of Ups1$_{complex}$. The increase in the RMSD of Ups1$_{complex}$ may suggest allosteric sites at the protein–protein interface, destabilizing some distant residues. This speculation was confirmed by root-mean-square-fluctuation (RMSF) profiles of Ups1 in these two simulations (Fig. 3d). As expected, Ups1$_{complex}$ had a lower residual fluctuation than Ups1$_{free}$ at the loop region (residues N43–N49) surrounded by Mdm35$_{complex}$ (Ups1$_{complex}$: ~1.5 Å, Ups1$_{free}$: ~2.5 Å), and had relatively higher RMSF in several distant segments (residues N28–H33, V66–L70, L100–G104, and K140–V163). The high RMSF was expected to influence the stable binding of Ups1 to the membrane.

Furthermore, Ups1$_{free}$ established stable protein-membrane interactions by inserting the F69 residue in the lipid bilayers

(Fig. 3c). Membrane insertion of F69 causes the center-of-mass of Ups1$_{free}$ to approach the membrane bilayer by 9 Å, thus potentially facilitating PA entry into the lipid-binding pocket of Ups1. A direct interaction between Ups1$_{free}$ and PA was observed after 800 ns (Supplementary Movies 1 & 3; Fig. 3e). The phosphate head of PA is surrounded by four positively charged residues H33, K61, K148, and K155, which were considered important for PA transfer[17,18]. Ups1$_{free}$ interacts with the lipid membrane through hydrophobic and positively charged residues in the L2-loop (N28–H33), $\alpha$2-loop (K61–R71), and C-terminal long $\alpha$3 helix (K140, R146, K154, M158, F162, K166, and R171). These membrane-bound structural elements contained those residues with high RMSF in the simulation of Ups1/Mdm35 complex system (Fig. 3c). The L2-loop is conserved, highlighting its importance for Ups1 function (Supplementary Fig. 9). The $\alpha$2-loop was considered a lid for the PA-binding pocket[21].

To determine whether Ups1 interacts with the membrane, as predicted through MD simulations, we performed the following experiments. First, we investigated whether the C-terminal $\alpha$3 helix of Ups1 is involved in membrane-binding. Ups1 exclusively binds to liposomes containing negatively charged phospholipids[8]. Our MD simulations revealed that Ups1 binds to the negatively charged membranes through some positively charged residues (K140, R146, K154, K166, and R171) in the $\alpha$3 helix. If these positively charged residues participate in membrane-binding, mutating them to negatively charged residues, e.g., Glu, would impair Ups1/Mdm35-mediated PA transfer. Therefore, we introduced various mutations at these residues and tested their potential for membrane-binding and PA transfer. The Ups1-2E mutant contains K140E and R146E double mutations. The Ups1-3E mutant contains K140E, R146E, and K154E triple mutations. The Ups1-4E mutant contains K140E, R146E, K154E, and K166E quadruple mutations. Ups1-2E alone completely eliminated the PA transfer activity of Ups1 (Fig. 3f). Because structural studies on Ups/Mdm35 excluded the potential interactions among these basic residues with Mdm35 or PA, the loss of PA transfer activity in these mutants may have resulted from the attenuation of Ups1-membrane interactions. This hypothesis was confirmed through liposome co-sedimentation assays (Fig. 3g, h). For these mutants, the amount of membrane-bound total Ups1 and Mdm35-complexed Ups1 significantly decreased. However, the amount of membrane-bound non-complexed Ups1 was significantly higher in these three mutants than that in the WT complex.

Furthermore, we investigated whether F69 is involved in membrane binding. F69 is located in the $\alpha$2-loop, which contains positively charged and hydrophobic residues (Supplementary Fig. 9). Its amino acid composition and structural similarity with the lipid exchange loop of PITP$\alpha$ facilitate its binding to negatively charged membranes. In this case, mutating F69 to a negatively charged residue Glu should impair the membrane-binding ability of Ups1/Mdm35, while mutating F69 to a hydrophobic residue, such as Leu, should have a limited effect. Liposome sedimentation assays confirmed our hypothesis. The overall membrane-binding ability of the F69E mutant was significantly impaired. Furthermore, the F69E mutation caused massive membrane accumulation of non-complexed Ups1 (Fig. 3i, j). For the F69L mutant, the amount of Mdm35-complexed Ups1 and non-complexed Ups1 on membranes was similar to that of the WT complex. We further mutated F69 into small hydrophobic (Ala), bulky uncharged polar (Tyr), and bulky hydrophobic (Trp) residues and measured their PA transfer activities (Fig. 3k). The PA transfer activity of the F69E mutant was almost abolished, while that mutants F69A and F69Y were considerably impaired. The PA transfer activities of mutants F69W and F69L almost remained unchanged, similar to the WT complex. Because F69 could be replaced with only residues with large hydrophobic

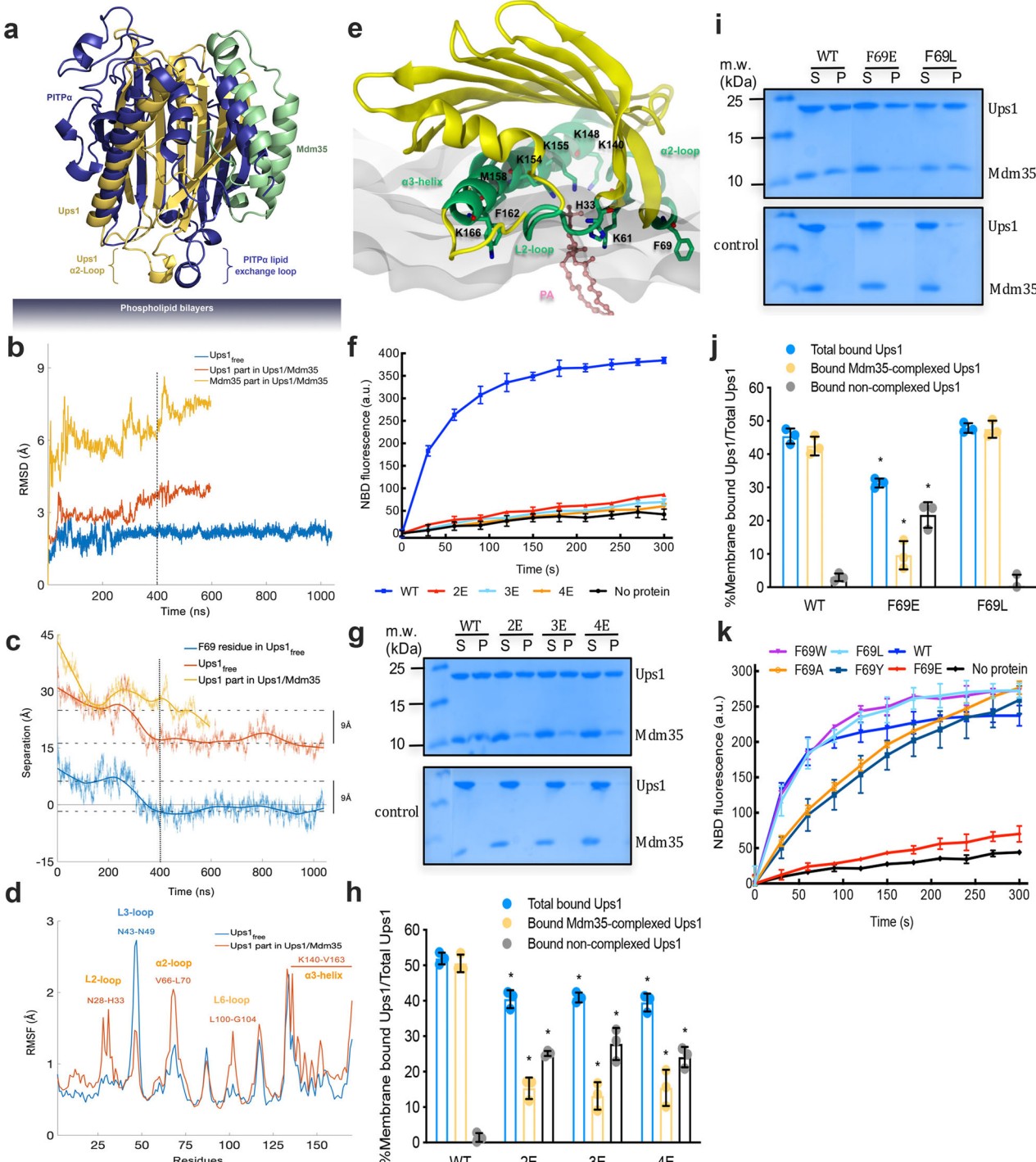

**Fig. 3 All-atom MD simulations of Ups1/Mdm35 or Ups1_free interactions with membranes. a** Initial Ups1/Mdm35 or Ups1_free orientations relative to the membrane in MD simulations. Ups1/Mdm35 is superposed with mouse apo-PITPα (PDB code, 1KCM). The identified lipid exchange loop in PITPα and the equivalent α2-loop in Ups1 are indicated. **b** RMSD profiles of Ups1/Mdm35 and Ups1_free simulations. **c** Time evolution profiles of separations between the lipid phosphate plane and the center of mass of Ups1_free, F69 in Ups1_free, and Ups1 part in Ups1/Mdm35 complex. Separation was defined as the vertical distance to the center of mass of lipid phosphorus atoms of the upper membrane bilayer. **d** Root-mean-square-fluctuation profiles of Ups1_free and the Ups1 part in the Ups1/Mdm35 complex. **e** A cross-sectional view of the interaction between Ups1_free and the lipid bilayer at 1023 ns. With the secondary structures involved in membrane-binding (α2-loop, L2-Loop, and α3 helix) colored in lime green, other parts of Ups1_free are colored in yellow. The hydrophobic and positively charged residues involved in membrane-binding are indicated. The PA molecule is shown as a ball-and-stick model and colored in yellow (carbon), red (oxygen), and orange (phosphorus). **f, k** PA transfer activities of WT Ups1/Mdm35 and the mutants. Traces indicate mean ± s.d. values of three independent experiments. **g, i** Representative Tricine-SDS-PAGE analysis of liposome co-sedimentation assays of WT Ups1/Mdm35 and the mutants from at least three independent experiments. Original uncropped gels are shown in Supplementary Fig. 14. **h, j** Quantification of Ups1 in different liposome-binding states related to **g** and **i**. See Fig. 2g for the quantification formula. Data represent three independent experiments and values are expressed as mean ± s.d. values. An independent samples t-test was performed using SPSS (version 23.0; n = 3, *p < 0.01).

side chains, it should be in a completely hydrophobic environment at least at some stage during the PA transfer process. A reasonable explanation is that this residue would be buried in the membrane during PA transfer, as predicted through MD simulations.

Furthermore, circular dichroism (CD) assays and one-dimensional (1D) $^1$H NMR assays revealed that these mutants and the WT complex shared similar spectra (Supplementary Fig. 10), indicating no remarkable conformational differences among them. Therefore, decreased PA transfer activities and altered membrane-binding behaviors of these mutants are not likely due to mutation-induced structural changes.

**Potential conformational changes on PA-binding**. To investigate how PA enters the PA-binding pocket of Ups1 and whether Ups1 elicits a conformational change before and after PA-binding, we compared the crystal structures of Ups1 in different states, including the MD-simulated membrane-bound (Ups1$_{free}$-1023 ns), the apo (PDB codes: 5JQM and 4YTW), the DLPA-bound (PDB code: 4YTX), and the DHPA-bound (PDB code: 6KYL) states. Furthermore, Watanabe et al. reported another state of Ups1 (PDB code: 4YTX, chain L), where Ups1 has a more open α2-loop but no PA in its lipid-binding pocket[17], called the X-state here.

After the five crystal structures were superimposed to the MD-simulated membrane-bound Ups1 structure, it was found that the conformations of the α2-loop and the N-terminal of the α3 helix (N-α3-helix) remarkably differed, even though their overall structures are similar (Fig. 4a). In the Ups1$_{free}$-1023 ns and apo-state structures (PDB codes: 5JQM, and 4YTW), the hydrophobic side chains of P63 and W65 in the α2-loop bind to the hydrophobic region formed by I137, V141, and W144 of the N-α3-helix, yielding a buried surface area of 499, 350, and 234 Å$^2$, which account for ~31%, 24%, and 15% of the total surface area of the α2-loop, respectively (Fig. 4b; Supplementary Table 1). However, in the PA-bound and X-state, both the α2-loop and the N-α3-helix swing outward, increasing the distance between the α2-loop and the N-α3-helix and eliminating their hydrophobic interactions (Fig. 4a).

To better compare the changes in proximity between the α2-loop and the N-α3-helix in these structures, we selected four adjacent residues (P63, W65, V141, and W144) in the α2-loop and N-α3-helix (Fig. 4b) and measured the distance between them (P63-V141, P63-W144, W65-V141, and W65-W144; Supplementary Table 1). The four distances were the smallest in the Ups1$_{free}$-1023 ns structure, followed by the apo, PA-bound, and the X states. Because both the α2-loop and the N-α3-helix are involved in membrane-binding, changes in their proximity lead to changes in the opening size on the Ups1-membrane interface (Fig. 4c). Membrane-anchored Ups1$_{free}$-1023 ns and the apo-state structures contain only a small hole on this interface, lined by residues H33, K61, K148, and K155, and might be involved in recognizing or binding to the PA head. For the two PA-bound Ups1 structures, the hole is enlarged into a cleft, probably because PA occupies this pocket. In the X-state, the cleft continues to expand to a wedge-shaped cavity. Despite no DLPA density in its PA-binding pocket, its structure might represent an intermediate state of PA entry and exit from the pocket; hence, a wider channel on the Ups1-membrane interface is required to accommodate the two long PA acyl chains. Interestingly, on re-analyzing the electron density of the X-state structure, we observed an additional density near the membrane interface, which could accommodate a DLPA molecule (Fig. 4d).

To investigate whether the hydrophobic interaction between the α2-loop and the N-α3-helix affects Ups1/Mdm35 function, we generated a W65A mutant and an I137A/V141A/W144A triple mutant. Both mutants had impaired PA transfer activity (Fig. 4e). We also constructed a W65A/F69A double mutant, where PA transfer activity of Ups1 was almost abolished (Fig. 4e).

Structural comparisons further revealed variations in the positions of F69 and W65 among different states (Fig. 4b). In PA-bound and X-state structures, W65 was inserted in the membrane to anchor Ups1. In the Ups1$_{free}$-1023 ns structure, F69 was inserted in the membrane. In apo-state structures, both F69 and W65 could not be inserted into the membrane. Conformational changes in the α2-loop and the N-α3-helix seemingly caused W65 and F69 to be alternatively inserted into the membrane during PA transfer (Supplementary Movie 4).

To test this hypothesis, we attempted to lock the position of W65 or F69 so that Ups1 membrane anchoring could be accomplished by only one of them (Fig. 4f). First, we designed the L70C/I103C mutant to lock F69. In apo structures (PDB codes: 5JQM and 4YTW), W65 interacts with I137, V141, and W144 and F69 is located above the membrane surface. The distance between the Cα atoms of L70 and I103 is 8.8 Å. If a disulfide bond is formed between L70 and I103 by mutating them to cysteine residues, the distance between their Cα atoms should be reduced to 5.5–5.7 Å. Consequently, the adjacent residue F69 should be further away from the membrane. W65 is less affected and can still be inserted into the membrane. Second, we designed the T64C/K140C mutant to lock W65 (Fig. 4f). By crosslinking T64C and K140C, the position of W65 is locked to interact with I137, V141, and W144, while F69 is almost unaffected and can still insert Ups1 into the membrane. We found the mutated Ups1 still complexed with Mdm35 (Supplementary Fig. 11), and each pair of the introduced cysteines spontaneously formed an intramolecular disulfide bond as expected (Fig. 4g; Supplementary Fig. 12). Although the two mutants could hardly transfer PA under non-reducing conditions, their PA transfer activities were restored after 10 mM DTT treatment (Fig. 4h). Hence, conformational changes (membrane insertion) in F69 and W65 are important for Ups1/Mdm35-mediated PA transfer.

**Regulation of PA transfer activity by pH**. Because the pH of the intermembrane space is greatly influenced by respiratory chain function[24], we speculated that pH potentially regulates Ups1/Mdm35-mediated PA transfer. After assessing Ups1/Mdm35-mediated PA transfer activity at different pH values, the optimum pH for Ups1/Mdm35 ranged 6.5–7.5 (Fig. 5a). At pH 6.0, the activity substantially decreased. At pH 5.5, the activity continued to decrease to half of that at pH 6.0.

Considering numerous basic residues on the membrane-binding interface of Ups1, we speculated that pH may affect the affinity of Ups1/Mdm35 to the membrane by changing the charges harbored by these residues. To verify this, we determined the electrostatic potentials and net charges of Ups1 molecules in different states at different pH values (Supplementary Table 2) using the PDB2PQR server[25]. Consequently, at the same pH, the amount of positive charges harbored by Ups1 is associated with its conformational states. However, regardless of the Ups1 conformational states, it contained the most positively charged residues at pH 5.5. Changes in electrostatic potentials at different pH values primarily occurred at the entrance to the PA-binding pocket (boxed regions in Fig. 5b). Apparently, at lower pH, Ups1 would have a stronger membrane-binding potential owing to increased amount of positively charged residues. This hypothesis was confirmed via the liposome co-sedimentation assays (Fig. 5c), where more non-complexed Ups1 accumulates on the membrane at lower pH.

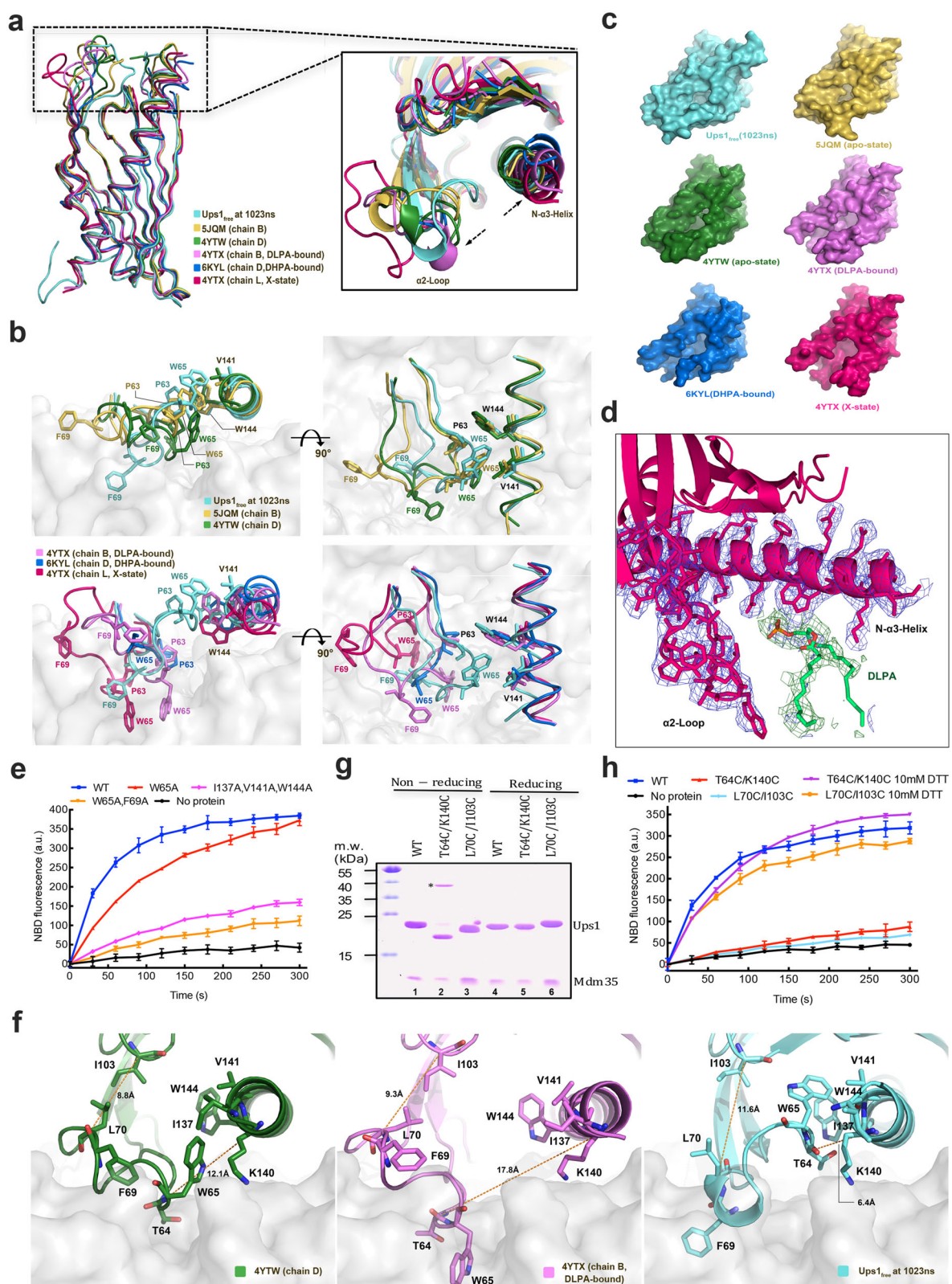

## Discussion

Although numerous crystal structures of Ups1/Mdm35 (or its homologs) in different states have been reported[17,18,21,23], the mechanism underlying PA transfer between mitochondrial membranes remains elusive. Here, we reported two new crystal structures of Ups1/Mdm35, performed all-atom simulations of Ups1-membane and Ups1/Mdm35-membrane, and various biochemical

and biophysical experiments, revealing how Ups1/Mdm35 interacts with the membrane and potential conformational changes of Ups1 during PA transfer.

Besides obtaining the authentic monomeric Ups1/Mdm35 structure and rectifying errors in previous crystallographic data, to the best of our knowledge, we resolved a novel complex structure of Ups1/Mdm35–DHPA, reporting the flexibility of the

**Fig. 4 Conformational changes in Ups1/Mdm35 during PA entry and binding. a** Superposition of six Ups1 structures with corresponding PDB codes. Magnified view shows the conformational changes in the α2-loop and the N-α3-helix. **b** Magnified view around the membrane-contacting surface of six Ups1 structures. The α2-loop and the N-α3-helix are denoted as ribbons. P63, W65, F69, V141, and W144 are indicated as stick models. **c** Potential membrane-contacting surfaces of six Ups1 structures. **d** Electron density map of a newly discovered DLPA molecule binding to Ups1 in the X-state (PDB code: 4YTX), which is colored in hot pink. The DLPA molecule is indicated with a ball-and-stick model and colored in green (carbon), red (oxygen), and orange (phosphorus). The electron density map of the α2-loop and the N-α3-helix is colored in marine. A simulated annealing $2F_o - F_c$ difference map was generated by omitting the DLPA molecule and is indicated with green meshes countered at 3.0 σ. **e** PA transfer activities of the WT Ups1/Mdm35 and the indicated mutants. Traces indicate mean ± s.d. values of three independent experiments. **f** Distances between the Cα atoms of L70 and I103 and between the Cα atoms of T64 and K140 in the corresponding Ups1 structures. **g** The T64C/K140C and L70C/I103C mutants were analyzed via SDS-PAGE with (reducing) or without (non-reducing) 10 mM DTT. The Ups1 bands of L70C/I103C and T64C/K140C mutants under non-reducing conditions migrate faster than those under DTT-treated reducing conditions. The raw gel image is included in Supplementary Fig. 14. The band marked with a black star in lane 2 is a Ups1 dimer formed through intermolecular disulfide crosslinking, identified through MS (Supplementary Fig. 12). We speculated it might be the classic domain-swapped Ups1 dimer in crystal structures. **h**. PA transfer activities of the WT Ups1/Mdm35 and the indicated mutants under reducing (pretreatment with 10 mM DTT for 3 min before analysis) or non-reducing conditions. Traces indicate mean ± s.d. values of three independent experiments.

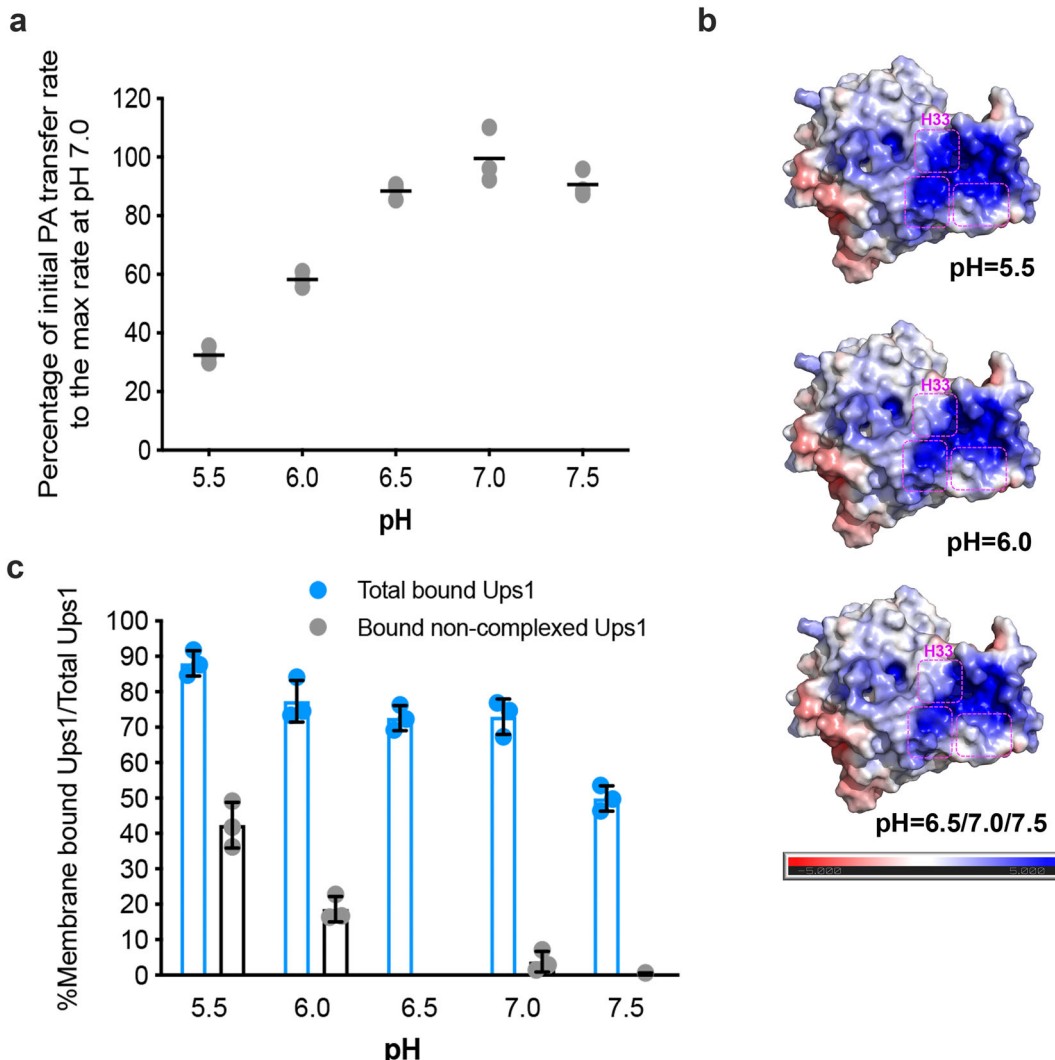

**Fig. 5 Effect of pH on the PA transfer activity of Ups1/Mdm35. a** PA transfer activities of Ups1/Mdm35 under different pH values. For each pH condition, fluorescence data from the first 60 s was used to determine the initial PA transfer rate. Traces indicate mean ± s.d. values of three independent experiments. **b** Electrostatic potential of the membrane-contacting surface of Ups1/Mdm35 under different pH values. Pink boxes with dotted lines highlight the regions with changed electrostatic potentials, one of which only contains residue H33. **c** Quantification of Ups1 in different liposome-binding states under different pH values. Ups1 and Mdm35 signals on gels were quantified and their values were substituted into the formula displayed in Fig. 2g. Data represent three independent experiments (gels are displayed in Supplementary Fig. 14) and values are expressed as the mean ± s.d. An independent samples t-test was performed using SPSS (version 23.0; $n = 3$, *$p < 0.01$).

αC-helix. These results led us to investigate how the Ups1–Mdm35 interaction is associated with PA transfer activity. Consequently, deletion of the αC-helix of Mdm35 considerably attenuates the PA transfer activity of Ups1.

The present MD simulations and subsequent mutagenesis studies revealed that not only does the α2-loop contribute to membrane-binding but also the L2-loop and α3-helix are involved in membrane interaction. We discovered that residues W65 and F69 of the α2-loop would be inserted into the membrane alternately during PA transfer. Although Miliara et al. suggested that the α3-helix and the Ω-loop (here, α2-loop) undergo subtle conformational changes, potentially contributing to phospholipid binding and release[26] and speculated that conformational differences in the Ω-loop (here, α2-loop) and α3-helix between PRELID1 and PRELID3b are potentially important for lipid specificity[23], no experimental data have supported this finding. Our disulfide-bridge experiments not only verified these conformational changes but also suggest a synergistic conformational change between the α2-loop and N-α3-helix.

Previously reported models presumed that Ups1 binds to the membrane on Mdm35 dissociation[18,21]. However, this assumption conflicts with the existing experimental data. First, the present and numerous previous studies[8,17,21] have reported that Mdm35 co-precipitates with liposomes in the presence of Ups1. Second, these data suggest that the amount of Ups1 co-precipitated with liposomes is always more than that of co-precipitated Mdm35. Considering that Mdm35 does not bind to the membrane itself[8], it is straightforward to deduce two states of membrane-bound Ups1, one is an Mdm35-associated state (Ups1/Mdm35) and the other is a Ups1-alone state (non-complexed state).

Once clarifying the two states of membrane-bound Ups1, we could determine the amount of these two states from our liposome co-precipitation experimental data and investigate how they influence PA transfer activity. We found that Ups1/Mdm35-mediated PA transfer decreases with an increase in the ratio of membrane-bound non-complexed Ups1 to total membrane-bound Ups1.

Our subsequent MD simulations suggested that only non-complexed Ups1 could stably interact with the membrane and mediate PA extraction instead of the Ups1/Mdm35 complex. Thus, non-complexed Ups1 could be an important transition state during Ups1/Mdm35-mediated PA transfer. Accordingly, we proposed a new model to explain the inverse correlation between PA transfer activity and the ratio of membrane-bound non-complexed Ups1. PA transfer activity of Ups1/Mdm35 depends on not only the rate of PA-binding to Ups1 but also Ups1 releasing rate from the membrane after PA-binding. Ups1 release from the membrane in solution needs Mdm35 re-association. Attenuation of membrane release of Ups1 would result in a subsequent reduction in PA transfer activity with an increased ratio of membrane-bound non-complexed Ups1.

The interactions of Mdm35 C-terminal truncation mutants with Ups1 were weakened, facilitating more Ups1 binding to membranes and attenuating the potential release of Ups1 from the membrane. Thus, an increased ratio of membrane-bound non-complexed Ups1, accompanying by the decreased PA transfer activity of these mutants, particularly for the mutant Δ (αC + C14), was observed.

For Ups1 α3-helix mutants, Ups1-membrane interactions were reduced, thus decreasing total membrane-bound Ups1 levels. However, increased ratios of non-complexed Ups1 are more remarkable, indicating that these mutations might allosterically attenuate the Ups1–Mdm35 interactions. Thus, the overall PA transfer activities of these mutants decreased considerably.

The pH-dependent PA transfer activity of Ups1/Mdm35 can also be well explained. The His residue (H33) of the L2-loop is located at the membrane-binding interface of Ups1 and presumably near the PA entry site. Considering a pKa of His residue of 6.0, a lower pH (5.5) would change this residue from deprotonated to protonated states. The increased positive charge at the interface would therefore enhance Ups1-membrane interactions, thus increasing the ratio of membrane-bound non-complexed Ups1 and attenuating PA transfer activity. However, we could not exclude another possibility that H33 protonation might also negatively affect PA entry into the Ups1 pocket.

Based on our study, here we propose a refreshed and detailed molecular model for Ups1/Mdm35-mediated PA transport (Fig. 6 and Supplementary Movie 4; see Supplementary Discussion and Supplementary Fig. 13 for details.). This model adequately explains the contradictory views regarding whether the α2-loop participates in membrane-binding. Watanabe et al.[17] reported the Δlid mutant, lacking the α2-loop (lacking residues L62-R71), has impaired PA transfer activity and can still bind to the membrane. They speculated that the α2-loop is not involved in membrane-binding but only serves as a gate for PA entry and exit. In our model, Ups1 binds to the membrane through an interface comprising the α2-loop, the C-terminal α3 helix, and the L2-loop. Without the α2-loop, Ups1 can still bind to the membrane via the other two structural elements. However, because conformational changes of α2-loop are critical for Ups1 function, in the absence of the α2-loop, Ups1/Mdm35-mediated PA transfer will inevitably be impaired.

Overall, our present study provides detailed insights into the mechanism underlying Ups1/Mdm35-mediated PA transport. The future studies investigating the structures of membrane-bound Ups1 with and without PA-binding are warranted.

## Methods

**Molecular cloning, protein expression, and purification**. To construct co-expression plasmids for N-terminal 6× His-tagged full-length Ups1 and full-length Mdm35, the corresponding genes were amplified through PCR from the genomic DNA of S. cerevisiae and cloned into pETDuet-1 (Novagen). To construct an N-terminal 6× His-tagged Ups1–Mdm35 fusion protein, the nucleotide sequence encoding the recognition sequence of PreScission protease (LEVLFQGP) was inserted between the C-terminal of Ups1 and the N-terminal of Mdm35 to generate an Ups1–Mdm35 fusion gene, which was also inserted in pETDuet-1. All constructs were sequenced to confirm their identities. The primers used in this study are listed in Supplementary Table 3. The genes of all the mutants in this study were synthesized by Beijing SYKM Gene Biotechnology Co., Ltd.

The target proteins were expressed in E. coli Trans B cells (TransGen Biotech) cultured in LB medium at 37 °C until the $OD_{600}$ approached 0.6–0.8. After supplementation of isopropyl-D-thiogalactoside (IPTG) to a final concentration of 0.1 mM, the cells were cultured at 30 °C for 5 h. Thereafter, cells were harvested and resuspended in buffer A (50 mM Tris-HCl, pH 8.0, 300 mM NaCl, 20 mM imidazole, and 1 mM PMSF) and then disrupted using a high-pressure homogenizer (JN-02C, JINBIO) at 120 MPa. The cell lysate was centrifuged using a JA25.5 rotor (Beckman Coulter) at 39,000×g for 40 min at 4 °C. The supernatant was loaded onto a column of Ni-chelated Sepharose 6 Fast Flow (GE Healthcare) pre-equilibrated with buffer A. The target protein was eluted using buffer B (50 mM Tris-HCl, pH 8.0, 300 mM NaCl, and 150 mM imidazole) and concentrated through ultrafiltration (30-kDa cutoff; Amicon Ultra). The eluate was further purified through size exclusion chromatography with Superdex 75 (10/300) column (GE Healthcare) in buffer C (20 mM Tris-HCl, pH 7.5 and 150 mM NaCl). The peak fractions were harvested, concentrated to approximately 20 mg mL$^{-1}$, and stored at −80 °C for further analysis. For the Ups1–Mdm35 fusion protein, the eluate was purified with Superdex 75 in buffer D (20 mM HEPES, pH 7.0, and 150 mM NaCl). Thereafter, the peak fractions were desalted and further purified through ion-exchange chromatography using a 1-mL Resource S column (GE Healthcare) in buffer E (20 mM HEPES, pH 7.0) and buffer F (20 mM HEPES, pH 7.0, and 1 M NaCl). Two peaks were obtained with a linear gradient of NaCl from 0 to 1 M in 20 column volumes. Fractions corresponding to the first peak were concentrated and stored at −80 °C for further experiments.

**Crystallization**. Crystallization was performed using the hanging-drop vapor diffusion method at 16 °C. To crystallize the Ups1/Mdm35 complex, a selenomethionine-derived Ups1/Mdm35 complex, or the Ups1–Mdm35 fusion

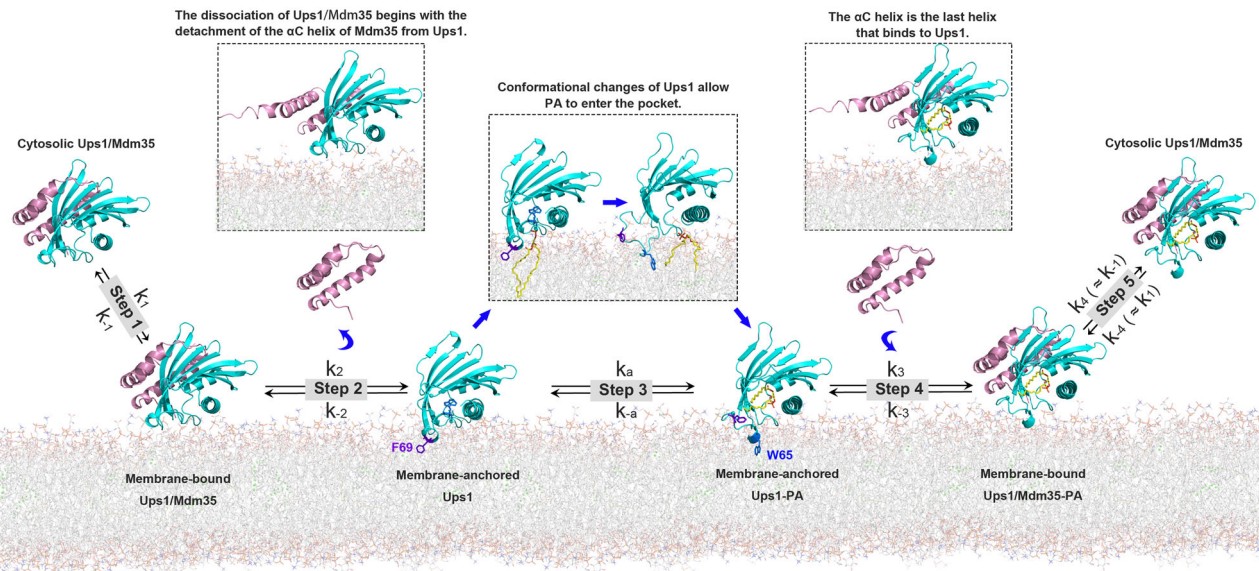

**Fig. 6 A working model for Ups1/Mdm35-mediated PA transport.** Step 1, Ups1/Mdm35 interacts with the membrane via the membrane-binding residues of Ups1. Step 2, the dissociation of Mdm35 from Ups1 leads to membrane insertion of F69, anchoring Ups1 into the membrane. Step 3, PA-binding results in a conformational change in Ups1, where W65 is inserted in the membrane instead of F69. Step 4, Mdm35 binds to Ups1 PA. Step 5, Ups1 PA/Mdm35 is released from the membrane to the aqueous environment.

protein, 1 μL of the protein solution (20 mg mL⁻¹), 1 μL of the reservoir solution (8% tacsimate, pH 5.0, 20% PEG3350), and 0.1 μL of 2 M sodium thiocyanate were mixed and equilibrated against 200 μL of the reservoir solution. To crystallize the Ups1/Mdm35–DHPA complex, purified Ups1/Mdm35 was incubated with 10-fold excess DHPA (830841 C, Avanti Polar Lipids) in buffer C for 30 min at 4 °C. Thereafter, the mixture was centrifuged at 17,000×g for 20 min at 4 °C, and 1 μL of the supernatant (20 mg mL⁻¹) and 1 μL of the reservoir solution (30–35% tascimate, pH 7.4, 0.1 M Bis-Tris propane, pH 7.5) were mixed and equilibrated against 200 μL of the reservoir solution.

**Data collection and structural determination.** Diffraction data were collected at 100 K at beamline BL17U and BL18U of Shanghai Synchrotron Radiation Facility and processed using an HKL2000 package[27]. Single-wavelength anomalous (SAD) diffraction data of the selenomethionine-substituted Ups1/Mdm35 crystal were curated at a 0.9791-Å wavelength. Selenium sites were determined using SHELXD[28]. Phases were determined and refined using SOLVE[29] and RESOLVE[30]. An initial model was built using COOT[31] and further refined with PHENIX[32]. The diffraction data of crystals of Ups1–Mdm35 fusion protein and Ups1/ Mdm35–DHPA were collected at a wavelength of 0.9791 and 0.9786-Å, respectively. Structures of Ups1/Mdm35–DHPA and Ups1–Mdm35 fusion protein were resolved through the molecular replacement method and further refined using PHENIX. Structure quality was validated using MolProbity[33]. Ramachandran analyzes of structures of selenomethionine-substituted Ups1/Mdm35, Ups1–Mdm35 fusion protein and Ups1/Mdm35–DHPA yielded the statistics of 98.2%/99.3%/96.3% residues in favored region and 1.8%/0.7%/3.3% in allowed region, respectively. The corresponding statistical data for structure refinement are summarized in Table 1.

**Liposome preparation.** Phospholipids 1,2-dioleoyl-sn-glycero-3-phosphocholine (DOPC; 850375C), 1,2-dioleoyl-sn-glycero-3-phosphoethanolamine (DOPE; 850725C), DOPA (840875C), and 18:1-12:0 1-oleoyl-2-(12-[(7-nitro-2-1,3-benzoxazol-4-yl) amino] dodecanoyl)-sn-glycero-3-phosphate, (NBD-PA; 810176C) were obtained from Avanti Polar Lipids. Rhodamine DHPE (L-1392) was purchased from ThermoFisher Scientific. Lipids in chloroform-containing stock solutions were mixed at the desired molar ratio, and the solvent was evaporated. The lipid film was hydrated in a suitable buffer, and the suspension was incubated at 37 °C for 30 min and then frozen in liquid nitrogen. Five freeze-thaw cycles were carried out and the thawed solution was extruded 21 times through a 0.1-μm filter, using a mini-extruder (Avanti Polar Lipids).

**Liposome co-sedimentation assay.** Liposomes (DOPC/DOPA = 80/20; 8 mM) were incubated with 20 μM Ups1/Mdm35 or its mutants in 50 μL of co-sedimentation buffer (20 mM Tris-HCl, pH 7.5, 150 mM NaCl and 1 mM EDTA) at 25 °C for 30 min. Negative control samples (without liposomes) were also prepared. The mixture was ultra-centrifuged at 250,000 g in a S140AT rotor (Doga Limited) on a micro-ultracentrifuge (Hitachi Koki himac CS-FNX series) at 25 °C for 30 min. The supernatants and pellets were subjected to 16.5% Tricine-SDS-

PAGE analysis. The stained gels from three parallel experiments were scanned and analyzed using Gel Doc EZ Gel Documentation System (Bio-Rad). The intensities of the pellets the negative controls were subtracted from the corresponding experimental data. The average statistical data of three sets of parallel experiments were used for final plotting and result analysis. Liposome co-sedimentation assays were also performed for Ups1/Mdm35 under different pH conditions. The buffers used here were the same as those used at different pH values in the PA transfer activity assay. Detailed quantitative data and statistics are summarized in Supplementary Data 1.

**PA transfer activity assay.** PA transfer activities of Ups1/Mdm35, Ups1–Mdm35 fusion protein, and the mutants were determined through the fluorescent dequenching assay[8,17]. Donor liposomes (6.25 μM; DOPC/DOPE/Rhodamine DHPE/18:1-12:0 NBD-PA = 50/40/2/8) were incubated with acceptor liposomes (25 μM; DOPC/DOPE/DOPA = 50/40/10) in the presence or absence of 20 nM purified protein in 2 mL of assay buffer (20 mM Tris-HCl, pH 7.5, 150 mM NaCl and 1 mM EDTA) at 25 °C. Furthermore, PA transfer activities of Ups1/Mdm35 under different pH conditions were measured. Assay buffers included MES buffer 5.5 (20 mM MES, pH 5.5, 150 mM NaCl and 1 mM EDTA), MES buffer 6.0 (20 mM MES, pH 6.0, 150 mM NaCl and 1 mM EDTA), PIPES buffer 6.5 (20 mM PIPES, pH 6.5, 150 mM NaCl and 1 mM EDTA), HEPES buffer 7.0 (20 mM HEPES, pH 7.0, 150 mM NaCl and 1 mM EDTA), and Tris buffer 7.5 (20 mM Tris-HCl, pH 7.5, 150 mM NaCl and 1 mM EDTA). NBD fluorescence was monitored using a fluorescence spectrometer (Hitachi F7000). The experimental raw data are described in Supplementary Data 2.

**Thermal shift assay.** A 5000× stock solution of Sypro orange dye (Life Technologies) was diluted 50-fold in water. Solutions of 10 μL of protein buffer (20 mM Tris-HCl, pH 7.5, 150 mM NaCl), 8 μL protein (2.0 mg mL⁻¹), and 2 μL of the diluted dye were placed in a 96-well PCR plate (Bio-Rad). The plate was then sealed with optical-quality sealing tape (Bio-Rad) and subjected to thermal treatment in a real-time PCR detection system (CFX96, Bio-Rad). The instrument was programed to equilibrate at 20 °C for 2 min, then increase the temperature by 0.5 °C per 30 s from 20 °C to 90 °C, and finally maintain the temperature at 90 °C for 2 min.

**CD determination.** CD spectra were recorded at ambient temperature, using the Chirascan plus CD spectrophotometer (Applied Photo-physics) with the sample in a quartz cuvette of 1-mm path length. Purified proteins with a final concentration of 0.2 mg mL⁻¹ in CD buffer (10 mM NaH₂PO₄, pH 7.5, 150 mM NaF, and 0.5 mM TCEP) were used for quantification. Data were obtained at 1 s per point with 1-nm bandwidth from 195 to 260 nm. After buffer background subtraction, data between 200 and 250 nm were used for analysis.

**1D ¹H NMR experiments.** Proteins (450 μL) in 20 mM PBS buffer were mixed with 50 μL of D₂O to prepare a 9:1 H₂O/D₂O NMR sample. All NMR data were acquired using a Agilent DD2 600 MHz spectrometer equipped with a z-axis

pulsed-field gradient triple resonance (1H, 13C, 15N) cold probe at 298 K. [1]H NMR spectra were recorded using the double pulsed-field gradient spin echo water suppression (DPFGSE) pulse sequence, 64 scans, and a 1.5-s relaxation delay. The acquired spectral width was 12 kHz and 16k complex points were obtained with a total acquisition time of 2.5 min. The final figure was generated on MestreNova.

**MD simulations**. Multicomponent lipid bilayers were constructed using CHARMM-GUI[34] with DOPC/DOPE/DOPI/TOCL2/DOPA/DOPG/DOPS (8:6:2:1:1:1:1). Ups1$_{free}$ and Ups1/Mdm35 complex systems were generated by placing the protein ~8 Å above the membrane surface. The systems were then solvated in explicit TIP3P[35] water molecules, with potassium and chloride ions to achieve a neutral 0.18 M ionic solvent, using the solvate and auto-ionize tools of VMD[36]. The MD simulations were carried out using NAMD 2.11 package[37] and the CHARMM[36] force field[38,39] with CMAP correction[40]. Electrostatic interactions were calculated using the particle mesh Ewald sum method[41] with a cutoff of 12 Å. All hydrogen-containing covalent bonds were constrained via the SHAKE algorithm[42] except that SETTLE algorithm[43] was used for water samples, thus facilitating an integration time step of 2 fs. Before production runs, the system was minimized in energy, heated to 310 K, and pre-equilibrated in the canonical ensemble while the protein backbone, and water oxygen atoms harmonically restrained with a spring constant of 10 kcal mol$^{-1}$ Å$^{-2}$. Simulations were then continued in the constant NPT ensemble (310 K and 1 atm). Langevin thermostats with a damping coefficient of 0.5 ps$^{-1}$ were used to regulate the system temperature. A Langevin-piston[44] barostat with a 2 ps piston period and a 2 ps damping time was used to regulate pressure. Constraints were next released in a step-wise manner (with the spring constant gradually decreasing from 10 to 0 kcal mol$^{-1}$ Å$^{-2}$) before the production runs. Total data spanning 600 and 1040 ns were generated for the Ups1/Mdm35 complex and Ups1$_{free}$ systems, respectively. Only those data obtained on reaching equilibrium were considered for further analysis. The root-mean-square-fluctuation (RMSF) profile was determined for each Cα atom of Ups1 over the final 200 ns for both simulations. The trajectories were aligned with their first frame, as the reference structure, before calculating the average positions of Cα atoms.

**Statistics and reproducibility**. Statistical analyses were performed using SPSS (version 23.0) using an independent samples *t*-test for triplicate experiments. All results are expressed as mean ± standard deviation (SD) values. Differences were considered statistically significant when $p < 0.01$(*). The tests were two-tailed. The actual *p*-value for each test can be found in the supplementary excel file provided herewith.

**Reporting summary**. Further information on research design is available in the Nature Research Reporting Summary linked to this article.

## Data availability
The coordinates and structural factors of crystal structures of selenomethionine-substituted UPS1/Mdm35, the fusion form and DLPA-bound state have been deposited in Protein Data Bank (accession codes: 5JQL, 5JQM, and 6KYL). The data required to generate statistics of liposome co-sedimentation experiments in Figs. 2i, 3h, j, and 5c are provided in Supplementary Data 1. The data required to generate plots of PA transfer assay in Figs. 2f, 3f, k, 4e, h, and 5a are provided in Supplementary Data 2. The data required to generate molecular dynamics plots in Fig. 3b–d are provided in Supplementary Data 3. Uncropped raw images of gels are shown in Supplementary Fig. 14. All other relevant data are available from the corresponding author upon request

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

## Acknowledgements

We would like to thank Ping Shan and Ruigang Su (F.S. lab) for their assistance in laboratory management. We are grateful to Qiangjun Zhou and Yun Zhu (F.S. lab) for their assistance in structure determination. We would like also to thank engineers, Jianhui Li, Xiang Ding, and Xuehui Liu from the Core Facility of Protein Sciences and Laboratory of Proteomics, Institute of Biophysics, for their assistance in fluorescence spectrometry, mass spectrometry analysis, and NMR analysis. This work was supported by grants from Ministry of Science and Technology of China (2017YFA0504700) to F.S., Natural Science Foundation of China (31771566) to Y.Z., Research Grants Council of Hong Kong (CityU 11306517) and NSFC/RGC Joint Research Scheme (N_CityU104/19) to J.F.

## Author contributions

F.S. and Y.Z. initiated and supervised the study. J.L. performed molecular cloning, protein expression, purification, crystallization, data collection, and structure determination. C.C. and J.F. performed molecular dynamics simulations. L.Y. performed mutagenesis experiments, biophysical assays, and the PA transfer activity assay. Y.Z. analyzed the data. Y.Z., J.F., and F.S. wrote the manuscript.

## Competing interests

The authors declare no competing interests.
