## [Peer Review File · Communications Biology]

Reviewers' comments:

Reviewer #1 (Remarks to the Author):

Since biosynthetic pathways of cellular phospholipids operate primarily in the ER membrane, but partly in the mitochondrial inner membrane, phospholipids need to be transported between the ER and mitochondrial outer membrane, and between the mitochondrial outer and inner membranes with the aid of still elusive lipid transport systems. Ups proteins and Mdm35 are one of the most well characterized lipid transport systems that function between the mitochondrial outer and inner membranes; the Ups1-Mdm35 complex transports phosphatidic acid (PA) from the outer membrane to the inner membrane within mitochondria, which is important for subsequent cardiolipin (CL) synthesis in the inner membrane. The mechanism of PA transfer by Ups1-Mdm35 is previously discussed well on the basis of their X-ray structures determined by several groups.

In the present study, Lu et al. determined the apo form and PA-bound form of Ups1-Mdm35 and discussed some details of the PA transfer mechanisms on the basis of the combination of the comparison of their structures with the previously reported structures, mutagenesis and lipid transfer and liposome binding assays, and molecular dynamics (MD) simulation. In particular, they stressed the role of the conformational change of the omega loop or $\alpha 2$ loop of Ups1 in the Ups1 PA transfer activity. However, the structure-based mechanism of binding of Ups proteins to the membrane was discussed previously, especially by the recent study that also used MD simulation (Miliara et al., Nat. Commun. 2019). Introduction of disulfide bridges to fix the positions of W65 and F69 is interesting, but the obtained results merely strengthen the previously established role of the omega loop in Ups functions.

The determined structure of the Ups1-Mdm35 apo form (5JQM) was the one without domain-swapping, but the other structure (5JQL) was in principle similar to the previously reported structures, but at lower resolution, so that there is no point of reporting this structure here and to PDB. The structure of the DHPA-bound form (5JQO) has a problem that the density of DHPA is too low, and the omit map is essential to validate the DHPA-bound structure. Thus, most of the findings reported here remained incremental to the previously reported works and fail to lead to novel insight into the structure-based mechanism of the Ups1 functions.

Specific points.

(1) Fig. 1e – Is the density map of the loop region of the Ups1-Mdm35 an omit map? If not, the omit map should be presented.

(2) Line 203-205 – The authors stated that only αA and αB helices of Mdm35 interact with Ups1. However according to the structure, αC helix interacts with neighboring Ups1 molecule, suggesting that the αC helix is domain-swapped with the αC helix of the neighboring Mdm35. Then, this statement may be misleading in that the αC helix is meant not to contribute to the interaction with Ups1.

(3) Fig. 2h, 3g, 3h, and 5c – Raw data of gel images are essential for validating the quantification.

(4) Liposome co-sedimentation assays cannot discriminate liposome-bound proteins and aggregated proteins. Therefore, a negative control without liposomes is essential to rule out potential artifacts arising from protein aggregation.

(5) Figs 2 and 3 -- Various Ups1 mutants showed only moderate changes in total binding of Ups1 to the membrane, but affected a lot the balance between membrane binding of Ups1 alone and of Ups1-

Mdm35. What does this mean? Since the previous model assumed that Ups1 binds to the membrane upon dissociation of Mdm35, explanation of this point is essential to propose a new model of Ups1 binding to the membrane.

(6) Paragraph starting from line 209 – To discuss the role of the C-terminal segment of Mdm35 in binding to Ups1, more direct methods like gel filtration assay is essential.

(7) Discussion depending too much on MD simulation is more suitable for specialized journals. The charge-changing mutations of the basic residues in the $\alpha 3$ helix of Ups1 affect lipid transfer and membrane binding activities of Ups1, yet these effects could be indirect, e.g. through a transient protein conformational change, and may not reflect the change in direct interactions between the basic residues in the $\alpha 3$ helix and the membrane. Direct contact of the proposed residues of Ups1 to the membrane can be tested by fluorescence measurement.

(8) In Fig. 4, the authors discussed structural differences among the different crystal structures assuming that they represent different states of Ups1 at work. However, their structural differences may have arisen from different crystallization conditions, packing effects, protein constructs etc.

(9) Fig. 4g – The stained bands of Ups1 look strange as compared with those of MW markers. Perhaps, they are overloaded? Asterisk should be explained.

(10) The letters in the figures are too small to read. The figures should be improved.

(11) In the discussion section, the authors state that the previous model assumes that the $\alpha 2$ -loop is not involved in membrane binding, but plays as a gate for PA entry and exit. However, the authors proposed here that the deletion of $\alpha 2$ -loop does not abolish membrane binding of Ups1 but affects PA extraction ability, thus impairing PA transfer activity of Ups1/Mdm35. I cannot see any substantial difference between the two interpretations.

(12) The authors' model and discussion in pages 22 and 23 are not based on the context of mitochondrial membranes since the inner membrane, where Ups1 releases PA, is enriched by cardiolipin as compared with the outer membrane. Therefore, this discussion lacks physiological relevance, and should be removed.

Reviewer #2 (Remarks to the Author):

In this manuscript the authors describe new insights of phosphatidic acid (PA) transfer learned from crystal structures, mutagenesis, and molecular dynamics (MD) simulations. The results advance our understanding of the molecular events that occur during PA transfer. The manuscript can be improved with addressing the following points and more quality controls in various aspects.

- The "unprecedented conformation" of the Ups1/Mdm35-DHPA structure was used as a basis to argue that the αC helix of Mdm35 detaches before dissociation from Ups1. However, this form could be an artifact of crystallization. It would be helpful to show/discuss the extent of crystal contacts. In fact, the results of the αC deletion functional assay seem to argue that this conformation may not be the relevant conformation, or at least not the dominant one, since deleting this region had a large effect on PA transfer. (If the conformation seen in the structure was the relevant one, then wouldn't deleting these residues make not much of a difference?)

- To substantiate any of the functional assays for the mutants (Fig 1b, 2f/h, 3f/g/h/i, 4e, 5a/c ...), some data showing the integrity/behavior of the protein should be presented. No gels are shown for any constructs except the one in Figure 4d, It should be straightforward for the authors to have some SEC chromatograms and gels included, even just in the supplement. Otherwise, it is unclear if functional defects in mutants are truly due to particular residues or just artifacts.
- Similarly, the authors have a nice schematic for how the liposome co-sedimentation assay works in Fig 2g, including what the raw data *would* look like and how to quantify it, but no raw data are actually shown.
- The high-resolution cutoff for the crystal structures are far too conservative (I/sigI 3-4.2, the one with DHPA not reported). More measured data should be included to improve the precision of the models.
- The descriptions of the types of interactions (lines 195-196) seem a bit loose... For example, one cannot say that N97 undergoes a hydrophobic interaction with the acyl chain of DHPA.
- Regarding the "Regulation of PA transfer activity by pH" section of Results:
 - Figure 5a has no error bars / info about statistics, and is not described in the Methods.
 - Figure 5b has no scale bar. As such, the take-home message feels a little unconvincing, with pretty minor differences between each subpanel. Better labeling or presentation would maybe make the point clearer.
- Figure 3c: It would be helpful to annotate features (α 2 loop, etc)
- Figure 3b and 3d: some data series are truncated at 600 ns compared to the others
- Figure 1e legend: Missing information on contour level
- Figure 4d legend: DLPA is green, not yellow
- The language could use quite a bit of attention/proofreading. For example, there are many incorrect words (such as multiple uses of "countered" instead of "contoured"), as well as incorrect grammar (such as starting a sentence with "To be noted that, [...]") and numerous typos (RMFS instead of RMSF, etc.)

Response to Reviewers

Reviewer #1 (Remarks to the Author):

Since biosynthetic pathways of cellular phospholipids operate primarily in the ER membrane, but partly in the mitochondrial inner membrane, phospholipids need to be transported between the ER and mitochondrial outer membrane, and between the mitochondrial outer and inner membranes with the aid of still elusive lipid transport systems. Ups proteins and Mdm35 are one of the most well characterized lipid transport systems that function between the mitochondrial outer and inner membranes; the Ups1-Mdm35 complex transports phosphatidic acid (PA) from the outer membrane to the inner membrane within mitochondria, which is important for subsequent cardiolipin (CL) synthesis in the inner membrane. The mechanism of PA transfer by Ups1-Mdm35 is previously discussed well on the basis of their X-ray structures determined by several groups.

Response #1:

We appreciate this reviewer for summarizing the phospholipids transfer from ER to mitochondria and pointing out the importance of studying PA transfer mechanism by Ups1-Mdm35 system. We believe our studies here made a big forward to understand the molecular mechanism by proposing a new model and discover the relationship between PA transfer activity and the ratio of non-complexed Ups1 on membrane.

In the present study, Lu et al. determined the apo form and PA-bound form of Ups1-Mdm35 and discussed some details of the PA transfer mechanisms on the basis of the combination of the comparison of their structures with the previously reported structures, mutagenesis and lipid transfer and liposome binding assays, and molecular dynamics (MD) simulation. In particular, they stressed the role of the conformational change of the omega loop or $\alpha 2$ loop of Ups1 in the Ups1 PA transfer activity. However, the structure-based mechanism of binding of Ups proteins to the membrane was discussed previously, especially by the recent study that also used MD simulation (Miliara et al., Nat. Commun. 2019). Introduction of disulfide bridges to fix the positions of W65 and F69 is interesting, but the obtained results merely strengthen the previously established role of the omega loop in Ups functions.

Response #2:

We would like to thank this reviewer for summarizing our work. However, first of all, we want to emphasize that the MD simulation by Miliara et al. is a kind of coarse-grain simulation, not all-atom simulation. To be noted, the residual fluctuation (RMSF) could be inaccurate in coarse-grained models (see Neri et al. PRL 95-218192-2005). Miliara et al. has stated in their paper that “Future work could exploit atomic resolution simulations using enhanced sampling methods in order to further characterize the dynamic Ω region in greater detail”, which is exactly we performed in this study. Our all-atom simulation provided atomistic details of protein-membrane interactions, refreshed our knowledge of the interface of Ups1 to interact with membrane, which not only includes $\alpha 2$ loop but also L2 loop and $\alpha 3$ helix.

Second, we disagree with this reviewer that our study merely strengthens the previously established results. Our work has the following novelties in terms of what the reviewer stated *“the structure-based mechanism of binding of Ups proteins to the membrane”*.

(1) Our study indicated that not only $\alpha 2$ loop contributes to membrane binding but also L2 loop and $\alpha 3$ helix are involved more importantly to interact with membrane.

(2) Although the conformational change of the $\alpha 2$ loop during PA transfer had been previously discussed, little is known about how this loop interacts with membrane and how its conformation responds to membrane binding and PA extraction. We, for the first time, discovered that residues W65 and F69 could be inserted into the membrane alternately, which was intensively evaluated by mutagenesis and biophysical assays, especially by the disulfide-bridges experiments.

(3) Although Miliara et al. suggested “ $\alpha 3$ helix and Ω loop undergo subtle conformational changes that would probably contribute to phospholipid binding and release” (Biochem.Soc.Trans. 2016) and speculated the conformational differences of Ω loop and $\alpha 3$ helix between PRELID1 and PRELID3b “are likely to be important for the lipid specificity of these lipid transfer proteins” (Nat. Commun. 2019), there were no experimental data to support. While, our disulfide-bridges experiments not only proved such conformational change but also suggested a synergistical conformational change between $\alpha 2$ loop and the N-terminal of $\alpha 3$ helix.

(4) The previous model assumed that Ups1 binds to the membrane upon dissociation of Mdm35, which cannot explain why Mdm35 is always present in Ups1-binding liposomes. We discovered the important role of the non-complexed Ups1 on membrane and proposed a new model that non-complexed Ups1 is an important transit for the success of PA transfer, and the equilibrium between non-complexed Ups1 and Ups1/Mdm35 complex on membrane affects the PA transfer rate and can be regulated by many factors including environmental pH.

Galileo said “all of the reasons must come from observation and experiments”. Previous experiments about the role of the omega loop ($\alpha 2$ loop) were limited and left speculations in understanding the function of Ups1/Mdm35. In our current work, we performed substantial experimental studies to provide an in-depth molecular picture of how Ups1/Mdm35 interacts with membrane and transfers PA.

The determined structure of the Ups1-Mdm35 apo form (5JQM) was the one without domain-swapping, but the other structure (5JQL) was in principle similar to the previously reported structures, but at lower resolution, so that there is no point of reporting this structure here and to PDB. The structure of the DHPA-bound form (5JQO) has a problem that the density of DHPA is too low, and the omit map is essential to validate the DHPA-bound structure. Thus, most of the findings reported here remained incremental to the previously reported works and fail to lead to novel insight into the structure-based mechanism of the Ups1 functions.

Response #3:

Before our work, there were two reported structures of Ups1-Mdm35 apo form with domain-swapping (PDB IDs, 4XHR and 4YTW). For the structure of 4XHR, the linker region (N134-I137) was incorrectly modelled previously and rebuilt in our work. And comparing the structure of 4YTW with our structure (5JQL), we will find that they came from different protein constructs that the full-length proteins of Ups1 and Mdm35 were used for 5JQL while the C-terminal five residues of both Ups1 and Mdm35 were truncated for 4YTW. In addition, their crystallization conditions are distinct and the unit cells of structures of 4YTW and 5JQL are remarkably different, yielding different number of molecules per asymmetric unit (see **Table R1**). Therefore, reporting

our structure 5JQL here and also to PDB are not merely a simple repeat to the previous work but providing additional useful information for future studies.

We would like to thank this reviewer for his/her critics upon the density of DHPA in the structure 5JQO. We have reprocessed the crystallographic data and updated the structural model with a new PDB ID 6KYL. The model of DHPA molecule was validated by both the omit annealing Fo-Fc map (see **Fig. 2b**) and the real-space correlation coefficients (see **Supplementary Fig. 5**).

Our work is not merely incremental to the previously reported work but more importantly provides novel understanding of how Ups1/Mdm35 works in substantial molecular details. Please see our **Response #2** to this reviewer.

Specific points

(1) Fig. 1e – Is the density map of the loop region of the Ups1-Mdm35 an omit map? If not, the omit map should be presented.

Response #4:

It is the 2Fo-Fc omit map. The figure legend has been revised to show this information.

(2) Line 203-205 – The authors stated that only \$\alpha A\$ and \$\alpha B\$ helices of Mdm35 interact with Ups1. However according to the structure, \$\alpha C\$ helix interacts with neighboring Ups1 molecule, suggesting that the \$\alpha C\$ helix is domain-swapped with the \$\alpha C\$ helix of the neighboring Mdm35. Then, this statement may be misleading in that the \$\alpha C\$ helix is meant not to contribute to the interaction with Ups1.

Response #5:

We appreciate this reviewer for his/her correction. Yes, we agree the αC helix is domain-swapped in the structure of Ups1/Mdm35-DHPA complex (PDB ID 6KYL in the revision). We have revised the text to eliminate the misleading and provided an additional figure to show the domain-swapped structure (see **Supplementary Fig. 3** in the revision).

(3) Fig. 2h, 3g, 3h, and 5c – Raw data of gel images are essential for validating the quantification.

Response #6:

Thanks a lot for this important suggestion. The representative gel images have been added into the corresponding figures and all the raw data of gel images have also been included in the supplementary material (see **Supplementary Fig. 14** in the revision).

(4) Liposome co-sedimentation assays cannot discriminate liposome-bound proteins and aggregated proteins. Therefore, a negative control without liposomes is essential to rule out potential artifacts arising from protein aggregation.

Response #7:

We appreciate this important comment. Indeed, we had performed the negative controls without liposomes (see **Supplementary Fig. 14** in the revision). We did not observe obvious protein aggregation without liposome. According to the reviewer's suggestion, we performed control subtraction before data statistics analysis (see **Supplementary File 1** in the revision). The relevant statistics figures (**Figs. 2i, 3h, 3k and 5c** in the revision) are also updated.

(5) Figs 2 and 3 -- Various Ups1 mutants showed only moderate changes in total binding of Ups1 to the membrane, but affected a lot the balance between membrane binding of Ups1 alone and of Ups1-Mdm35. What does this mean? Since the previous model assumed that Ups1 binds to the membrane upon dissociation of Mdm35, explanation of this point is essential to propose a new model of Ups1 binding to the membrane.

Response #8:

We appreciate this reviewer to emphasize this point, which is actually one of novelties of our work. The previously reported models assumed Ups1 binds to the membrane upon dissociation of Mdm35 (see Fig. 5B in Miliara et al. EMBO Rep. 2015, and Fig. EV9 in Yu et al. EMBO Rep. 2015). However, this assumption conflicts with the existing experimental data including ours. First, there have been many published data (see Fig. S6 and S16 in Connerth et al. Science 2012, Fig. 5d in Watanabe et al. Nat. Commun. 2015, and Fig. 4c in Miliara et al. EMBO Rep. 2015) as well as our data (**Figs. 2h, 3g and 3i** in the revision) to demonstrate Mdm35 co-precipitates with the liposome in the presence of Ups1. Second, these data also suggest the amount of Ups1 co-precipitated with liposomes is always more than that of Mdm35

co-precipitated. Considering Mdm35 does not bind to membrane itself (see Fig. S6 in Connerth et al. Science 2012), it is straightforward to deduce that there are two states of Ups1 bound to membrane, one is Mdm35-associated state (Ups1/Mdm35) and the other is Ups1-alone state (non-complexed state).

With the knowledge of the two states of Ups1 on membrane, we could calculate the amount of these two states from our liposome co-precipitation experimental data and correlate with the PA transfer assay. We found **the PA transfer activity of Ups1/Mdm35 decreases as the ratio of membrane-bound non-complexed Ups1 to total Ups1 increases**. This is one of novel discoveries from our study.

Our subsequent MD simulations suggested that only the non-complexed Ups1 could form a stable interaction with membrane and be responsible for PA extraction instead of Ups1/Mdm35 complex. Based on this observation, we could propose a new model to explain the inverse correlation between PA transfer activity and the ratio of non-complexed Ups1 on the membrane. PA transfer activity of Ups1/Mdm35 depends on not only the rate of PA binding to Ups1 but also the releasing rate of Ups1 from membrane after PA binding. The releasing of Ups1 from membrane to solution needs re-association of Mdm35. If the releasing rate of Ups1 from membrane is attenuated, it will result a subsequent decrease of PA transfer activity with an increased ratio of non-complexed Ups1 on the membrane.

For the C-terminal truncation mutants of Mdm35 (**Fig. 2**), their interactions with Ups1 are weakened, allowing more Ups1 binding to membrane but also attenuating the releasing ability of Ups1 from membrane. Then the increased ratio of non-complexed Ups1 on the membrane was observed, which correlated with the decreased PA transfer activity of the mutants, especially for the mutant $\Delta(\alpha C+C14)$.

For the mutations at $\alpha 3$ helix (**Fig. 3**), the interaction between Ups1 and membrane is reduced, resulting a decrease of total bound Ups1 on membrane. However, the increased ratios of non-complexed Ups1 are more remarkable, indicating these mutations also attenuate allosterically the interaction between Mdm35 and Ups1 on membrane. Thus, the overall PA transfer activities of these mutants were largely decreased.

The discussion has been further revised to incorporate the above explanation and make a better description of our proposed model.

(6) Paragraph starting from line 209 – To discuss the role of the C-terminal segment of Mdm35 in binding to Ups1, more direct methods like gel filtration assay is essential.

Response #9:

Thanks for this nice suggestion. During revision process, we performed gel filtration chromatography for all mutants mentioned in our manuscript and the results have been added into supplementary materials (see **Supplementary Fig.11** in the revision). Truncations of C-terminal segment of Mdm35 did not alter the formation of Ups1/Mdm35 complex from gel filtration. The relevant text has been revised to incorporate this new information.

(7) Discussion depending too much on MD simulation is more suitable for specialized journals. The charge-changing mutations of the basic residues in the $\alpha 3$ helix of Ups1 affect lipid transfer and membrane binding activities of Ups1, yet these effects could be indirect, e.g. through a transient protein conformational change, and may not reflect the change in direct interactions between the basic residues in the $\alpha 3$ helix and the membrane. Direct contact of the proposed residues of Ups1 to the membrane can be tested by fluorescence measurement.

Response #10:

We performed substantial MD simulations. However, our discussions were not only based on MD simulations but also coming from large amount of experimental data including crystal structures, mutagenesis and biophysical/biochemical assays.

To rule out the possibility that the mutations of $\alpha 3$ helix would induce a transient conformational change, we further measured Circular Dichroism (CD) spectrums and one-dimensional ^1H NMR spectrums on the wild-type Ups1/Mdm35 and the charge-changing mutants (see **Supplementary Figure 10** in the revision). The spectrums of the mutants and wild type are exactly similar and no obvious conformational changes could be observed.

The total bound Ups1 on membrane was reduced significantly (**Fig. 3h**) for these charge-changing mutants, which is a strong evidence for direct interaction. However, as we response above (#8), the charge-changing mutations would also attenuate allosterically the interaction between Mdm35

and Ups1 on membrane, affecting the ratio of non-complexed Ups1 on membrane more significantly.

We did not perform fluorescence measurement since we consider the fluorescence labeling of lipid or the target residues might influence the interaction between Ups1 and membrane and the resulting data might not be directly explained.

(8) In Fig. 4, the authors discussed structural differences among the different crystal structures assuming that they represent different states of Ups1 at work. However, their structural differences may have arisen from different crystallization conditions, packing effects, protein constructs etc.

Response #11:

We appreciate this critic. We agree different crystallization conditions, packing effects and protein constructs would yield structural differences, which might not be functional relevant. However, in our work, the assumption that different crystal structures represent different states of Ups1 at work was actually verified by disulfide bridge mutagenesis. Thus, although structural differences we discussed were from comparisons of crystal structures, our subsequent experimental assays assured their functional relevance.

The relevant text has been revised to avoid this potential misunderstanding.

(9) Fig. 4g – The stained bands of Ups1 look strange as compared with those of MW markers. Perhaps, they are overloaded? Asterisk should be explained.

Response #12:

The protein marker we used in Fig.4g was Thermo Scientific Page Ruler Prestained Protein Ladder #26616, its color was different from normal Coomassie R-250 staining bands. The raw image of the gel has been shown in **Supplementary Figure 15** in the revision.

We believe the asterisk band represents the mutant Ups1-T64C/K140C cross-linked by intramolecular disulfide bond (see **Supplementary Figure 12**), and have explained in the revised legend.

(10) The letters in the figures are too small to read. The figures should be improved.

Response #13:

Thanks for this suggestion. The figures have been improved with bigger letters.

(11) In the discussion section, the authors state that the previous model assumes that the α 2-loop is not involved in membrane binding, but plays as a gate for PA entry and exit. However, the authors proposed here that the deletion of α 2-loop does not abolish membrane binding of Ups1 but affects PA extraction ability, thus impairing PA transfer activity of Ups1/Mdm35. I cannot see any substantial difference between the two interpretations.

Response #14:

Both interpretations agree α 2-loop plays the role in PA extraction and release, while their substantial difference is whether the α 2-loop participates in membrane binding. In the previous model, α 2-loop is not involved in membrane binding. However, according to our model, Ups1 binds to the membrane through several structural features, including α 2-loop, L2-loop and α 3 helix. Therefore, after deleting α 2-loop, Ups1 can still bind to membrane via other two structural elements.

(12) The authors' model and discussion in pages 22 and 23 are not based on the context of mitochondrial membranes since the inner membrane, where Ups1 releases PA, is enriched by cardiolipin as compared with the outer membrane. Therefore, this discussion lacks physiological relevance, and should be removed.

Response #15:

We agree in the physiological condition, the net result is that Ups1 extracts PA from outer membrane and releases to inner membrane. However, we should be aware that thermodynamics process is microscopically reversible. The net direction of PA transfer from outer to inner membrane is driven by the high concentration of PA in outer membrane compared to inner membrane. The role of Ups1/Mdm35 is just to accelerate the rate of PA transfer but could not determine the direction of PA transfer. This is the physical thermodynamics picture that our analysis and discussion is based on.

We have revised our kinetics model to avoid any potential misunderstanding.

Reaction (a) can be considered as the PA extraction process on the outer membrane while reaction (b) as the PA release process on the inner membrane. Considering the different lipid composition between outer and inner membrane, the corresponding reaction constants in (a) and (b) would not be same.

However, in our liposome co-sedimentation experiments, the liposomes play both donor membrane and acceptor membrane. With this circumstance, the reactions (a) and (b) are actually becoming same. The ratios we measured in the liposome co-sedimentation experiments represent the equilibrium state of the reaction (a)/(b). With the further assumption that k_4/k_{-4} is approximately equal to k_{-1}/k_1 , we could derive and calculate the equilibrium constant K_1 (see **Supplementary Fig. 13**), which helps us to make a deeper understanding of how Ups1/Mdm35 interacts with the membrane.

Overall, our theoretical model was based on the consideration of the physiological condition, while the subsequent derivation was simplified on the basis of the liposome co-sedimentation experiments. Our theoretical calculation is self-consistent with our experimental data. Thus this part of discussion is very important to describe a deep picture of how Ups1/Mdm35 works in molecular details, and could not be removed.

Reviewer #2 (Remarks to the Author):

In this manuscript, the authors describe new insights of phosphatidic acid (PA) transfer learned from crystal structures, mutagenesis, and molecular dynamics (MD) simulations. The results advance our understanding of the molecular events that occur during PA transfer. The manuscript can be improved with addressing the following points and more quality controls in various aspects.

1. The "unprecedented conformation" of the Ups1/Mdm35-DHPA structure was used as a basis to argue that the α C helix of Mdm35 detaches before dissociation from Ups1. However, this form could be an artifact of crystallization. It would be helpful to show/discuss the extent of crystal contacts. In fact, the results of the α C deletion functional assay seem to argue that this conformation may not be the relevant conformation, or at least not the dominant one, since deleting this region had a large effect on PA transfer. (If the conformation seen in the structure was the relevant one, then wouldn't deleting these residues make not much of a difference?)

Response #1:

We would like to thank this reviewer for this valuable critic. As we response to Reviewer #1 (see our response #5), after analyzing crystal contacts, we found Ups1/Mdm35-DHPA also forms a domain-swapping structure with α C helix swapped (see **Supplementary Fig. 3** in the revision). The α C helix of Mdm35 interacts with a neighboring Ups1 molecule with the interaction area of $\sim 375 \text{ \AA}^2$, which is smaller than the area of $\sim 431 \text{ \AA}^2$ in the normal apo structure.

Although the "unprecedented conformation" of Mdm35 could be an artifact of crystallization, this conformation reflects the flexibility of α C helix, which allows us to hypothesize α C helix is relevant to dissociation/association of Mdm35. Because the α A and α B helices of Mdm35 are cross-linked by two intramolecular disulfide bonds, they interact with Ups1 as a unified entity. The interaction area between α A- α B helices of Mdm35 and Ups1 is $\sim 837 \text{ \AA}^2$, much larger than that between α C helix and Ups1. The α C helix has the weaker interaction with Ups1 and greatest flexibility. Thus, α C helix is most likely the element that the dissociation begins with (or the association ends with).

The reviewer might not be aware of the relationship between Mdm35 dissociation/association and PA transfer activity, which is actually the novel

point of our study (see our response #8 to Reviewer #1). Deleting α C helix had a large effect on PA transfer is exactly because the weakened interaction between Mdm35 and Ups1 attenuates the ability of Mdm35 to release Ups1-PA from the membrane, thus resulting the decrease of PA transfer activity. The weakened interaction between the deletion mutant of Mdm35 and Ups1 was further confirmed by thermal shift assay (see **Supplementary Fig. 8b** in the revision).

The manuscript has been revised to include a description of the above crystal contact analysis (see **Supplementary Fig. 3** in the revision) and more analysis about the relationship between the ratio of non-complexed Ups1 on the membrane and the PA transfer activity.

2. To substantiate any of the functional assays for the mutants (Fig 1b, 2f/h, 3f/g/h/l, 4e, 5a/c ...), some data showing the integrity/behavior of the protein should be presented. No gels are shown for any constructs except the one in Figure 4d, it should be straightforward for the authors to have some SEC chromatograms and gels included, even just in the supplement. Otherwise, it is unclear if functional defects in mutants are truly due to particular residues or just artifacts.

Response #2:

Thanks for this important suggestion. We have put gel images of the liposome co-sedimentation assays in the main figures (**Figs. 2h, 3i and 3g**). All the raw gel images have been provided as **Supplementary Fig. 14**. According to the reviewer's suggestion, we managed to perform SEC chromatography (**Supplementary Figs. 8a and 11**), CD spectrometry (**Supplementary Fig. 10a**) and ¹H-NMR spectrometry (**Supplementary Fig. 10b**) to confirm the integrity/behaviors of the mutants are same with the wild type.

*3. Similarly, the authors have a nice schematic for how the liposome co-sedimentation assay works in Fig 2g, including what the raw data *would* look like and how to quantify it, but no raw data are actually shown.*

Response #3:

For the raw data, please see our response #2 above. We have included **Supplementary File 1** in the revision to describe the details of gel quantification and ratio computation.

4. The high-resolution cutoff for the crystal structures are far too conservative (I/sigI 3-4.2, the one with DHPA not reported). More measured data should be included to improve the precision of the models.

Response #4:

We appreciate this valuable comment. We turned back to check the crystallographic datasets again and found we could not include more measured data due to the current crystal-to-detector distance and the limited detector area. The current resolution has already reached the edge of the detector for either SeMet Ups1/Mdm35 dataset or the fusion Ups1-Mdm35 dataset. When including more measured data at the corner, the completeness of the high resolution shell would be significantly decreased. Thus, we would like to keep the current cutoffs and statistics for these two datasets in Supplementary Table 1.

For the DHPA bound structure, according to the comment of this reviewer, we have re-processed the data and re-refined the structure with the updated statistics in Supplementary Table 1. The updated structure has been submitted to PDB with the accession code of 6KYL. The relevant structural validation report has been provided in supplementary material.

5. The descriptions of the types of interactions (lines 195-196) seem a bit loose... For example, one cannot say that N97 undergoes a hydrophobic interaction with the acyl chain of DHPA.

Response #5:

The descriptions have been revised accordingly.

6. Regarding the "Regulation of PA transfer activity by pH" section of Results: Figure 5a has no error bars / info about statistics, and is not described in the Methods.

Response #6:

We have added error bars in **Fig. 5a** and made a description in the Methods.

7. Figure 5b has no scale bar. As such, the take-home message feels a little unconvincing, with pretty minor differences between each subpanel. Better labeling or presentation would maybe make the point clearer.

Response #7:

We have added scale bar and marked the differences in each subpanel.

8. Figure 3c: It would be helpful to annotate features (\$\alpha\$ 2 loop, etc.)

Response #8:

We have labeled the features accordingly (see the revised **Fig. 3d**).

9. Figure 3b and 3d: some data series are truncated at 600 ns compared to the others.

Response #9:

The data series of Ups1/Mdm35 were not truncated at 600 ns. The total simulation time was 600 ns and the convergence reached. Since Ups1/Mdm35 does not form a stable interaction with membrane and we focused on the interaction between Ups1 alone and the membrane, to save computing resources, we did not continue the simulation after observing the convergence.

10. Figure 1e legend: Missing information on contour level

Response #10:

The contour level has been added into the legend.

11. Figure 4d legend: DLPA is green, not yellow

Response #11:

Corrected.

12. The language could use quite a bit of attention/proofreading. For example, there are many incorrect words (such as multiple uses of "countered" instead of "contoured"), as well as incorrect grammar (such as starting a sentence with "To be noted that, [...]") and numerous typos (RMFS instead of RMSF, etc.)

Response #12:

We have asked an English language edition service (Wallace Academic Editing) to help proofread the manuscript and correct grammar mistakes.

Table R1. Crystallographic parameter comparisons between two crystal structures of Ups1/Mdm35 in apo form

PDB ID		4YTW	5JQL
Resolution (Å)		1.4	2.9
Space group		P1211	P1211
Unit cell	Length (Å)	a=42.8, b=71.7, c=87.6	a=83.3, b=130.7, c=84.3
	Angle (°)	$\alpha=90.0$, $\beta=95.0$, $\gamma=90.0$	$\alpha=90.0$, $\beta=103.1$, $\gamma=90.0$
Number of molecules per asymmetric unit		2	6
Crystallization condition		0.2 M ammonium citrate pH 7.0, 20% PEG 3350	8% Tacsimate, pH5.0, 20% PEG 3350, 200 mM Sodium thiocyanate

REVIEWERS' COMMENTS:

Reviewer #1 (Remarks to the Author):

This is a revised version of the previously submitted manuscript. The authors responded to most of my concerns and comments, and made clear what the points of this work are. The new and appealing points the authors are making are summarized below, yet I still feel that the results shown here are incremental and concerning very details of the previously established model of the PA transfer mechanism.

(1) As the authors stated in their reply and described more clearly in the revised manuscript, one of the appealing points of this work could be the presence as well as distinct roles of free Ups1 and the Ups1-Mdm35 complex that are bound to membranes in PA transfer. However, in this revised model, Mdm35 facilitates the release of Ups1 from the membrane, and it is the Mdm35-free form that extracts PA from the membrane, which does not change the previously established model of the PA transfer between the membranes by Ups1 in cooperation with Mdm35 significantly.

(2) Another seemingly important point of this work could be the model that conversion of the Ups1-alone state to the Mdm35-associated state of Ups1 facilitates the PA transfer by promoting dissociation of Ups1-Mdm35 complex from the membrane (lines 469-479 in the revised manuscript). The authors used Mdm35 truncation mutant, $\Delta(\alpha C+C14)$, which is weak in association with Ups1 and exhibits lower PA extraction activity from liposomes, to draw this conclusion. However, the thermal shift assays (Supplementary Fig. 8b) showed that the Sypro orange binding to the $\Delta(\alpha C+C14)$ mutant is different from that to other Mdm35 derivatives including WT Mdm35, suggesting that the $\Delta(\alpha C+C14)$ mutant likely exposes hydrophobic regions differently in addition to its lower stability of the complex between the $\Delta(\alpha C+C14)$ mutant and Ups1. Therefore, interpretation of the observation on the $\Delta(\alpha C+C14)$ may not be so simple. In relation to this, the authors had better see, in the PA transfer and liposome binding assays using the $\Delta(\alpha C+C14)$ mutant (Figs. 2f and 2h), the competition between WT Mdm35 and $\Delta(\alpha C+C14)$ mutant in re-binding to free Ups1 on the membrane; the authors can test the effect of the addition of free WT Mdm35 or free $\Delta(\alpha C+C14)$ mutant to the Ups1- $\Delta(\alpha C+C14)$ complex in these assays, which will strengthen the authors' claim of the 'new' model.

(3) Minor points.

Line 999: "intramolecular disulfide" should read "intermolecular disulfide".

Fig. 2h: Lanes for $\Delta C5$ and $\Delta C14$ may have been reversed.

Reviewer #2 (Remarks to the Author):

The authors made a thorough revision addressing all of my concerns. The manuscript is substantially improved and suitable for publication.

Response to Reviewers

Reviewer #1 (Remarks to the Author):

This is a revised version of the previously submitted manuscript. The authors responded to most of my concerns and comments, and made clear what the points of this work are. The new and appealing points the authors are making are summarized below, yet I still feel that the results shown here are incremental and concerning very details of the previously established model of the PA transfer mechanism.

(1) As the authors stated in their reply and described more clearly in the revised manuscript, one of the appealing points of this work could be the presence as well as distinct roles of free Ups1 and the Ups1-Mdm35 complex that are bound to membranes in PA transfer. However, in this revised model, Mdm35 facilitates the release of Ups1 from the membrane, and it is the Mdm35-free form that extracts PA from the membrane, which does not change the previously established model of the PA transfer between the membranes by Ups1 in cooperation with Mdm35 significantly.

Response #1:

We appreciate that our last round of response is acknowledged by this reviewer. The previously established model of the PA transfer between membranes by Ups1 does include the cooperation of Mdm35. Our new model does not make a conflict at this point. However, the previous model is rough without a clear picture of how Ups1 interacts with membrane. It is also not clear whether Mdm35 could still attach to Ups1 after Ups binds to membrane. Our new model provides a clarified picture of the detailed process during PA transfer including membrane association, Ups1 bound to membrane, Mdm35 disassociation, conformational change of $\alpha 2$ loop, PA binding, and re-association of Mdm35. The role of $\alpha 2$ loop is also re-visited and corrected in our model. The discovery of the relationship between PA transfer activity and the ratio of non-complexed Ups1 on membrane is very important to understand the molecular mechanism. Thus, we think our model is not just an incremental but a big forward step to the previously proposed model.

(2) Another seemingly important point of this work could be the model that conversion of the Ups1-alone state to the Mdm35-associated state of Ups1 facilitates the PA transfer by promoting dissociation of Ups1-Mdm35 complex from the membrane (lines 469-479 in the revised manuscript). The authors used Mdm35 truncation mutant, $\Delta(\alpha C+C14)$, which is weak in association with Ups1 and exhibits lower PA extraction activity from liposomes, to draw this conclusion. However, the thermal shift assays (Supplementary Fig. 8b) showed that the Sypro orange binding to the $\Delta(\alpha C+C14)$ mutant is different from that to other Mdm35 derivatives including WT Mdm35, suggesting that the $\Delta(\alpha C+C14)$ mutant likely exposes hydrophobic regions differently in addition to its lower stability of the complex between the $\Delta(\alpha C+C14)$ mutant and Ups1. Therefore, interpretation of the observation on the $\Delta(\alpha C+C14)$ may not be so simple. In relation to this, the authors had better see, in the PA transfer and liposome binding assays using the $\Delta(\alpha C+C14)$ mutant (Figs. 2f and 2h), the competition between WT Mdm35 and $\Delta(\alpha C+C14)$ mutant in re-binding to free Ups1 on the membrane; the authors can test the effect of the addition of free WT Mdm35 or free $\Delta(\alpha C+C14)$ mutant to the Ups1- $\Delta(\alpha C+C14)$ complex in these assays, which will strengthen the authors' claim of the 'new' model.

Response #2:

We appreciate this reviewer for his/her rigorousness. The thermal shift assays (**Supplementary Fig. 8b**) were performed on the sample of Ups1/Mdm35 complexes not Mdm35 proteins alone. Thus, the high fluorescence signal of Sypro orange bound to Ups1/ $\Delta(\alpha C+C14)$ would not be due to the exposed hydrophobic regions of $\Delta(\alpha C+C14)$ but in most probability the hydrophobic region of Ups1 that would interact with αC helix of WT Mdm35 and is exposed in Ups1/ $\Delta(\alpha C+C14)$ complex.

In addition, it is not the level of fluorescence signal but the critical temperature that can reflect the thermal stability of the protein directly. Therefore, we calculated the first derivative of the original thermal shift fluorescence curve to compute the critical temperature T_m for protein denaturing (see **Supplementary Fig. 8c** in the revision). It is clear that the critical temperatures of Ups1/ $\Delta C14$ and Ups1/ $\Delta(\alpha C+C14)$ are much lower than that of WT and Ups1/ $\Delta C5$. Thus, the lower stability of the complex Ups1/ $\Delta(\alpha C+C14)$ in comparison with WT is clear.

The new assay suggested by this reviewer would be very interesting but not quite necessary at the current point. We will perform these experiments in our future studies.

(3) Minor points.

Line 999: “intramolecular disulfide” should read “intermolecular disulfide”.

Fig. 2h: Lanes for $\Delta C5$ and $\Delta C14$ may have been reversed.

Response #3:

We appreciate this reviewer for his/her great carefulness. These typos have been corrected in the final revision.

Reviewer #2 (Remarks to the Author):

The authors made a thorough revision addressing all of my concerns. The manuscript is substantially improved and suitable for publication.

Response #1:

Many thanks, we appreciate.